# AN EMPIRICAL STUDY AND THEORETICAL EXPLANATION ON TASK-LEVEL MODEL-MERGING COLLAPSE

## ABSTRACT

Model merging unifies independently fine-tuned LLMs from the same base, enabling reuse and integration of parallel development efforts without retraining. However, in practice we observe that merging does not always succeed: certain combinations of task-specialist models suffer from catastrophic performance degradation after merging. We refer to this failure mode as merging collapse. Intuitively, collapse arises when the learned representations or parameter adjustments for different tasks are fundamentally incompatible, so that merging forces destructive interference rather than synergy. In this paper, we identify and characterize the phenomenon of task-level merging collapse, where certain task combinations consistently trigger huge performance degradation across all merging methods. Through extensive experiments and statistical analysis, we demonstrate that representational incompatibility between tasks is strongly correlated with merging collapse, while parameter-space conflict metrics show minimal correlation, challenging conventional wisdom in model merging literature. We provide a theoretical explanation on this phenomenon through rate-distortion theory with a dimension-dependent bound, establishing fundamental limits on task mergeability regardless of methodology.

## 1 INTRODUCTION

Large language models (LLMs) Brown (2020); Bommasani et al. (2021); Achiam et al. (2023); Grattafiori et al. (2024); Yang et al. (2024a); Guo et al. (2025) have demonstrated remarkable success in numerous downstream tasks from natural language processing Wang et al. (2018); Hendrycks et al. (2020) to code generation Lu et al. (2021); Chen et al. (2021); Jain et al. (2024); Li et al. (2024). While LLMs achieve strong general performance Achiam et al. (2023); Grattafiori et al. (2024); Yang et al. (2024a); Guo et al. (2025), adapting them to specific domains through fine-tuning remains computationally intensive Yadav et al. (2024a), particularly when multiple specialized variants are required for different tasks or applications Wolf et al. (2020); Yadav et al. (2024a). This challenge has spurred interests in parameter-efficient methods Hu et al. (2022); Yadav et al. (2024a) to combine and adapt pretrained models without exhaustive retraining.

Among emerging solutions, model merging Zhang et al. (2019); Wortsman et al. (2022); Ilharco et al. (2023); Yadav et al. (2024a); Yu et al. (2024); Lu et al. (2024) has shown particular promise as a computationally efficient approach to combine multiple fine-tuned LLMs derived from the same base model. By merging model parameters directly, model merging aims to consolidate specialized capabilities into a single unified model while avoiding the costs of additional training. By directly combining model parameters, merging leverages the widely observed linear mode connectivity Frankle et al. (2020); Zhou et al. (2024) between fine-tuned variants to create multi-capability models. Task arithmetic methods Zhang et al. (2019); Wortsman et al. (2022); Ilharco et al. (2023) and their recent extensions Yadav et al. (2024a); Yu et al. (2024); Lu et al. (2024) have shown particular effectiveness for specific task combinations. However, a critical question remains underexplored: What fundamental limitations govern which tasks can be successfully merged without model-merging collapse (defined in Section 2)?

In this paper, we identify and characterize the phenomenon of task-level merging collapse. Through comprehensive empirical analysis spanning five state-of-the-art merging methods Zhang et al. (2019); Wortsman et al. (2022); Ilharco et al. (2023); Yadav et al. (2024a); Yu et al. (2024, diverse LLM

architectures Grattafiori et al. (2024); Yang et al. (2024a), and fine-tuning approaches Howard & Ruder (2018); Hu et al. (2022), we observe a consistent and alarming pattern: model merging can catastrophically fail when combining certain task combinations, despite each individual model performing well in isolation. Contrary to the prevailing focus on parameter conflicts Yadav et al. (2024a), our statistical analysis reveals that parameter-space metrics show minimal correlation.

To explain this phenomenon and find better correlated metric, we introduce the first theoretical framework for analyzing model merging through the lens of rate-distortion theory Berger (2003). Under the Locally Modified Components (LMC) assumption, we prove a fundamental bound relating merging distortion to the geometry of hidden representations. Our dimension-dependent theorem establishes that for representations in $R^d$, no convex merging method can achieve distortion below $\Delta^2 \cdot \frac{d}{2(d+1)}$, where $\Delta$ is the diameter of task-specific representation clusters. Our theoretical analysis aligns with the empirical findings, where hidden-state representational incompatibility at the task level is strongly correlated with merging collapse, providing a better understanding of what drives merging degradation and actionable insights for guiding task selection in improving model merging.

In summary, our main findings and contributions are as follows:

- **Task-level representational incompatibility drives merging collapse.** We demonstrate that merging failure is fundamentally determined by task compatibility rather than methodology, with certain task combinations consistently causing catastrophic performance degradation across all merging approaches. Through controlled experiments, we establish that task-level representational conflicts strongly predict merging failure, while parameter-space conflict metrics show minimal correlation, challenging conventional wisdom in model merging literature.

- **Dimension-aware theoretical framework.** We formalize task-level merging collapse using information theory, proving that for representations in $\mathbb{R}^d$, the minimum achievable hidden-state distortion is bounded by $\Delta^2 \cdot \frac{d}{2(d+1)}$ (Theorem 1). This dimension-dependent bound establishes precise quantitative limits on task mergeability, connecting empirical collapse to fundamental information-theoretic constraints.

- **Empirical validation of theoretical explanation.** We validate our theoretical predictions through extensive experiments across diverse model architectures, tasks, and merging methods. Our results confirm that representation incompatibility statistically correlates with merging collapse, with observed merging collapse following the theoretical explanations. These findings provide a principled explanation for previously puzzling cases of merging collapse.

## 2 PRELIMINARY

The development lifecycle of LLMs Bommasani et al. (2021) typically follows a two-stage paradigm: pre-training and fine-tuning. While the pre-trained model develops a wide-ranging understanding, the representations learned during pre-training are not optimized for any specific application, necessitating further fine-tuning on task-specific data with targeted objectives.

Let $M_\theta$ denote a Foundation Model (FM) with trainable parameters $\theta$, which can represent either the entire model for full fine-tuning or a subset of parameters for parameter-efficient fine-tuning techniques Hu et al. (2022); Liu et al. (2022); Houlsby et al. (2019). In collaborative FM development environments, multiple teams often independently fine-tune the same base model for different capabilities or on different data distributions Touvron et al. (2023), resulting in a collection of specialized models. Consider a set of tasks $\mathcal{T} = \{t_1, t_2, ..., t_n\}$. The based FM $M_{\theta_0}$ is fine-tuned on each task $t_i$, resulting in fine-tuned model $M_{\theta_i}$ with updated parameters $\theta_i$. The distributed development creates the need for effective approaches to consolidate these independently evolved models.

**Model merging.** Building upon the independent fine-tuning efforts described above, there arises a crucial need to integrate the strengths of multiple specialized models. Model merging Wortsman et al. (2022); Zhang et al. (2019); Ilharco et al. (2023); Chitale et al. (2023); Ortiz-Jimenez et al. (2024); Yadav et al. (2024a); Yu et al. (2024); Yang et al. (2024b) refers to the process of combining multiple independently fine-tuned models with their parameter updates into a unified model that

preserves the capabilities of its constituents. Given the updated parameters $\{\theta_1, ..., \theta_n\}$ obtained from fine-tuning the common base model $M_{\theta_0}$, the goal of model merging is to produce a consolidated model $M_{\theta_{merged}}$ that preserves the task-specific knowledge from each fine-tuned model.

In order to achieve merging task-specific knowledge, many existing works Hendel et al. (2023); Hojel et al. (2024); Ilharco et al. (2023); Chitale et al. (2023); Ortiz-Jimenez et al. (2024) leverage *Task vectors*, which provides a concise representation of the model update resulting from fine-tuning on a specific task. Formally, let $\theta_0 \in \mathbb{R}^d$ denote the pre-trained model weights, and $\theta_t \in \mathbb{R}^d$ the weights after fine-tuning on task $t$. The task vector is defined as $\tau_t = \theta_t - \theta_0$, capturing both the direction and magnitude of parameter changes induced by fine-tuning. In other words, $\tau_t$ encodes the element-wise modifications made to the model parameters for adapting to task $t$. Furthermore, task vectors enable a straightforward approach to model merging: by averaging the task vectors from $n$ tasks, one can construct a merged model as $\theta_{merged} = \theta_0 + (\tau_1 + \tau_2 + \cdots + \tau_n)/n$.

**Parameter update conflicts.** When merging models fine-tuned on different tasks, conflicts often arise within the parameter updates Yadav et al. (2024a); Yang et al. (2024b). A direct and intuitive form of conflict occurs when the elements in task vectors for different tasks have opposite signs, implying that the optimal parameter update for one task may be detrimental to another Yadav et al. (2024a). However, even in the absence of sign differences, significant disparities in the magnitudes of updates can also lead to suboptimal performance in the merged model.

Based on established literature Huang et al. (2024); Yadav et al. (2024a); Ilharco et al. (2023), we focus on these four metrics to provide a comprehensive view of parameter update conflicts.

- **Parameter magnitude change ratio** is the ratio between the sum of the difference of each parameter of the same position and the sum of two models' task vectors' magnitude.
- **Parameter sign change ratio** is the ratio between the number of positions where two models' task vector not sharing a same sign and the total number of parameters positions.
- **Conflicting parameter magnitude change ratio** is the ratio between the difference of parameters at positions where two models' task vector not sharing a same sign and the sum of two models' task vectors' magnitude.
- **Average cosine similarity between pair's task vectors** stands for the average cosine similarity between two model's parameter vectors.

**Merging collapse**. Similar to model collapse training on synthesis data Shumailov et al. (2024), we introduce the concept of merging collapse as phenomenon when distinct fine-tuned models cannot be successfully combined by a given model merging technique while preserving their original capabilities. Formally, let $P(\theta_i, T_i)$ denote the performance of model $M_{\theta_i}$ on task $T_i$, we quantify merging collapse using the **merging loss** $-100\% \leq L(T_i) \leq 0\%$ on each task $T_i$:

$$L(T_i) = (\frac{\mathrm{P}(\theta_{\mathrm{merged}}, T_i)}{\mathrm{P}(\theta_i, T_i)} - 1) \times 100\% \tag{1}$$

As for multiple tasks, we further define the average merging loss across all tasks as the arithmetic mean. In the analysis that follows, we omit the percentage sign "%" for simplicity.

## 3 EMPIRICAL INVESTIGATION OF TASK-LEVEL MODEL-MERGING COLLAPSE

While previous work has demonstrated that model merging can effectively combine knowledge from multiple fine-tuned models, these studies typically focus on specific merging techniques Yadav et al. (2024a); Yu et al. (2024) or specific model architecture Yadav et al. (2024b). A comprehensive understanding of when and why model merging succeeds or fail across different conditions remains an open challenge. To bridge the gap, we conduct a comprehensive empirical study answering the following research questions:

- **RQ1:** How consistently does merging collapse occur across different merging techniques?
- **RQ2:** To what extent is merging collapse method-dependent or task-dependent? Do different merging techniques exhibit similar collapse patterns when confronted with the same task combinations?

- **RQ3:** What factor best correlates with merging compatibility? In particular, how do representation-space metrics (capturing task-level conflicts) compare to parameter-space metrics (measuring weight update conflicts) in explaining merging collapse?

## 3.1 STUDY SETUP

**Models and datasets.** We randomly select 64 checkpoints from Lots-of-LoRAs Collection Brüel-Gabrielsson et al. (2024) to cover a diverse range of tasks, which are finetuned with LoRA Hu et al. (2022) from Mistral-7B Jiang et al. (2023). We also choose eight tasks COLA, MNLI, MRPC, QNLI, QQP, RTE, SST-2 and WNLI from the GLUE dataset Wang et al. (2018), which are widely used by previous work Yadav et al. (2024a); Lu et al. (2024) for evaluating model merging techniques. We fine-tune 5 models including Llama3.2-3B Grattafiori et al. (2024), Llama3.1-8B Grattafiori et al. (2024), Qwen2.5-3B, 7B, and 14B Yang et al. (2024a), T5-Base, T5-Large, and T5-XL Raffel et al. (2023). These models cover different scales (from 300M to 14B), architectures (decoder-only and encoder-decoder), and training approaches (instruction tuning and task-specific fine-tuning) to improve the representativeness of our findings. We fine-tune these models on the eight GLUE tasks, resulting in 64 model checkpoints. Details of model checkpoints and fine-tuning settings can be found in our Appendix D.

**Model merging techniques.** We experiment with five state-of-the-art model merging techniques including **Linear Averaging (LA)** Zhang et al. (2019); Wortsman et al. (2022), **Task Arithmetic (TA)** Ilharco et al. (2023); Chitale et al. (2023); Ortiz-Jimenez et al. (2024), **TIES** Yadav et al. (2024a), **DARE** Yu et al. (2024), and **SLERP** Freeden & Törnig (1981); Goddard et al. (2024) implemented in MergeKit Goddard et al. (2024).

**Merging settings.** For GLUE tasks, we merge every 8 fine-tuned checkpoints from the same base model with all the five model merging techniques. For model checkpoints from Lots-of-LoRAs, we generate 25 task groups (a)∼(y)of merging tasks, with each group containing 8 randomly selected checkpoints. We reproduce linear averaging, task arithmetic and TIES on these checkpoint groups.

**Evaluation metrics.** In merging experiments of Lots-of-LoRAs, we measure the performance with rougeL Lin (2004) as Lots-of-LoRAs mentioned. In merging experiments of checkpoints fine-tuned on GLUE dataset, we measure the performance by classification accuracy. We use the merging loss defined by Equation 1 in Section 2 to quantitatively measure the model mergeability.

## 3.2 RQ1: MODEL MERGING COLLAPSE

First, we conduct comprehensive experiments on GLUE tasks. Table 1 presents the merging loss range of existing merging techniques on GLUE tasks from which we have two major observations.

**All models suffer from severe mergeability issues in model merging.** Regardless of model architecture, model size, or merging technique, every model examined exhibits substantial merging losses when combining multiple models simultaneously. Even the best-performing combinations show double-digit merging losses, with values reaching to -32.8 across different configurations for GLUE tasks. This universal degradation demonstrates that current merging techniques fundamentally break down when scaled to practical multi-model scenarios that more closely resemble real-world deployment needs.

Table 1: Merging Losses on GLUE tasks across Different Merging Techniques and Models.

| Model | Merging Techniques | | | | |
|-------|------|------|------|------|-------|
|       | **LA** | **TA** | **TIES** | **DARE** | **SLERP** |
| Llama3.2-3B | $17.3 \pm 7.5$ | $17.6 \pm 7.8$ | $21.8 \pm 16.9$ | $57.0 \pm 27.5$ | $17.5 \pm 8.0$ |
| Llama3.1-8B | $22.0 \pm 22.2$ | $22.0 \pm 22.2$ | $31.7 \pm 19.9$ | $66.1 \pm 27.9$ | $22.2 \pm 23.0$ |
| Qwen2.5-3B | $13.7 \pm 15.1$ | $13.6 \pm 15.2$ | $13.3 \pm 13.0$ | $27.1 \pm 16.8$ | $13.1 \pm 14.7$ |
| Qwen2.5-7B | $17.4 \pm 23.3$ | $17.0 \pm 22.2$ | $26.1 \pm 24.3$ | $62.6 \pm 24.6$ | $17.6 \pm 22.6$ |
| Qwen2.5-14B | $19.0 \pm 26.3$ | $19.1 \pm 26.9$ | $18.6 \pm 25.7$ | $31.9 \pm 27.8$ | $18.9 \pm 26.3$ |
| T5-Base | $29.0 \pm 19.5$ | $29.5 \pm 20.5$ | $31.6 \pm 21.4$ | $44.5 \pm 13.1$ | $27.7 \pm 22.5$ |
| T5-Large | $26.3 \pm 24.9$ | $34.4 \pm 27.0$ | $26.8 \pm 23.5$ | $28.2 \pm 27.2$ | $28.2 \pm 27.2$ |
| T5-XL | $20.0 \pm 18.8$ | $19.4 \pm 18.2$ | $19.3 \pm 18.7$ | $21.2 \pm 17.3$ | $25.7 \pm 23.3$ |

**No model merging technique overcomes the mergeability issue in model merging.** While certain techniques perform marginally better in specific contexts, none successfully mitigates the fundamental mergeability problem at scale. This suggests that mergeability limitations represent

Table 2: Effectiveness of different merging methods on NLP Tasks with Qwen2.5 Models.

| Model | Merging | NLP Tasks | | | | | | | | | | | | | | | | |
|---|---|---|---|---|---|---|---|---|---|---|---|---|---|---|---|---|---|
| | | COLA | | MNLI | | MRPC | | QNLI | | QQP | | RTE | | SST-2 | | WNLI | |
| | | FT | M(Δ) | FT | M(Δ) | FT | M(Δ) | FT | M(Δ) | FT | M(Δ) | FT | M(Δ) | FT | M(Δ) | FT | M(Δ) |
| 3B | LA | 82.7 | 81.0 (-2.1) | 88.4 | 86.2 (-2.4) | 89.5 | 75.5 (-15.6) | 92.6 | 85.0 (-8.2) | 84.2 | 62.6 (-25.7) | 90.3 | 85.2 (-5.6) | 95.4 | 93.7 (-1.8) | 73.2 | 38.0 (-48.1) |
| | TA | 82.7 | 81.2 (-1.9) | 88.4 | 86.2 (-2.5) | 89.5 | 75.5 (-15.6) | 92.6 | 85.1 (-8.1) | 84.2 | 62.6 (-25.7) | 90.3 | 85.6 (-5.2) | 95.4 | 93.6 (-1.9) | 73.2 | 38.0 (-48.1) |
| | TIES | 82.7 | 79.2 (-4.3) | 88.4 | 87.8 (-0.7) | 89.5 | 83.6 (-6.6) | 92.6 | 67.8 (-26.9) | 84.2 | 63.2 (-24.9) | 90.3 | 87.7 (-2.8) | 95.4 | 91.9 (-3.7) | 73.2 | 46.5 (-36.5) |
| | DARE | 82.7 | 70.4 (-14.9) | 88.4 | 35.4 (-60.0) | 89.5 | 82.6 (-7.7) | 92.6 | 50.6 (-45.4) | 84.2 | 63.2 (-24.9) | 90.3 | 80.9 (-10.4) | 95.4 | 73.7 (-22.7) | 73.2 | 50.7 (-30.8) |
| | SLERP | 82.7 | 81.7 (-1.3) | 88.4 | 86.2 (-2.4) | 89.5 | 76.0 (-15.1) | 92.6 | 85.4 (-7.8) | 84.2 | 62.6 (-25.6) | 90.3 | 85.9 (-4.8) | 95.4 | 93.7 (-1.8) | 73.2 | 39.4 (-46.2) |
| 7B | LA | 86.1 | 82.5 (-4.2) | 90.8 | 86.1 (-5.1) | 88.0 | 74.0 (-15.9) | 95.3 | 88.3 (-7.4) | 88.0 | 63.2 (-28.2) | 89.9 | 88.1 (-2.0) | 96.0 | 94.5 (-1.6) | 78.9 | 19.7 (-75.0) |
| | TA | 86.1 | 82.9 (-3.7) | 90.8 | 86.1 (-5.1) | 88.0 | 74.0 (-15.9) | 95.3 | 88.1 (-7.6) | 88.0 | 63.2 (-28.2) | 89.9 | 87.4 (-2.8) | 96.0 | 94.4 (-1.7) | 78.9 | 22.5 (-71.4) |
| | TIES | 86.1 | 62.2 (-27.7) | 90.8 | 88.2 (-2.8) | 88.0 | 55.4 (-37.0) | 95.3 | 64.4 (-32.5) | 88.0 | 63.2 (-28.2) | 89.9 | 89.5 (-0.4) | 96.0 | 94.4 (-1.7) | 78.9 | 16.9 (-78.6) |
| | DARE | 86.1 | 0.0 (-100.0) | 90.8 | 27.0 (-70.2) | 88.0 | 31.6 (-64.1) | 95.3 | 50.5 (-47.0) | 88.0 | 63.2 (-28.2) | 89.9 | 47.7 (-47.0) | 96.0 | 0.0 (-100.0) | 78.9 | 43.7 (-44.6) |
| | SLERP | 86.1 | 82.6 (-4.0) | 90.8 | 87.0 (-3.8) | 88.0 | 73.3 (-16.7) | 95.3 | 88.4 (-7.3) | 88.0 | 63.2 (-28.2) | 89.9 | 85.9 (-4.4) | 96.0 | 94.4 (-1.7) | 78.9 | 21.1 (-73.2) |
| 14B | LA | 88.1 | 81.5 (-7.5) | 91.9 | 90.1 (-1.9) | 90.4 | 78.2 (-13.6) | 95.9 | 89.0 (-7.2) | 88.7 | 62.4 (-29.7) | 93.9 | 88.8 (-5.4) | 97.1 | 95.1 (-2.1) | 84.5 | 12.7 (-85.0) |
| | TA | 88.1 | 81.4 (-7.6) | 91.9 | 90.2 (-1.9) | 90.4 | 78.7 (-13.0) | 95.9 | 89.0 (-7.3) | 88.7 | 62.4 (-29.7) | 93.9 | 89.2 (-5.0) | 97.1 | 95.2 (-2.0) | 84.5 | 11.3 (-86.7) |
| | TIES | 88.1 | 81.6 (-7.4) | 91.9 | 88.2 (-4.1) | 90.4 | 81.6 (-9.8) | 95.9 | 81.5 (-15.1) | 88.7 | 70.8 (-20.2) | 93.9 | 88.1 (-6.2) | 97.1 | 96.1 (-1.1) | 84.5 | 12.7 (-85.0) |
| | DARE | 88.1 | 28.4 (-67.8) | 91.9 | 57.8 (-37.1) | 90.4 | 85.5 (-5.4) | 95.9 | 50.5 (-47.3) | 88.7 | 76.0 (-14.3) | 93.9 | 92.1 (-1.9) | 97.1 | 92.4 (-4.8) | 84.5 | 19.7 (-76.7) |
| | SLERP | 88.1 | 81.4 (-7.6) | 91.9 | 90.2 (-1.9) | 90.4 | 78.9 (-12.7) | 95.9 | 89.2 (-7.0) | 88.7 | 62.4 (-29.6) | 93.9 | 88.4 (-5.8) | 97.1 | 95.2 (-2.0) | 84.5 | 12.7 (-85.0) |

* FT: Fine-tuned model performance; $M(\Delta)$: Performance after merging and the negative number of merging loss following Equation 1.

an inherent challenge to scaled model merging rather than merely a technical limitation of current approaches. The DARE technique exhibits a steepest increase in loss, which is found out to be due to its failure on certain task as previous work Deng et al. (2025) shows. Other techniques like LA, TA, TIES, and SLERP show more moderate but still significant loss, typically ranging from 10-25%.

To further validate the generalization of model-merging collapse across different models and merging techniques, we conduct comprehensive experiments on Lots-of-LoRAs task groups. Out of the 25 Lots-of-LoRAs task groups we merge, 2/3 of them experience a worst performance loss of more than 30% when merged, and only one group maintains performance within 10% of the original models, demonstrating that severe merging collapse is the norm rather than the exception. We have detailed data presentation in Table 24~48 and a visualization in Appendix Figure 6.

> **Answer to RQ1:** Model-merging collapse do exist for all merging techniques. Even the best-performing merging techniques show significant degradation. This suggests mergeability limitations are inherent to scaled model merging rather than technical shortcomings of specific technique.

### 3.3 RQ2: Method vs. Task Dependence in Merging Collapse

To investigate whether the revealed merging collapse in RQ1 is due to the imperfection of existing merging methods or due to the inherent incompatibility of model checkpoints, in Table 2 we present a detailed merging results with Qwen2.5 models. It can be observed from Table 2 that there exists some tasks, such as MRPC and WNLI, seem to suffer severe merging loss across all merging techniques, indicating the choice of merging technique has little impact over performance in certain tasks.

Table 3: P-values of One-way ANOVA F-tests Maxwell et al. (2017) for Task-level and Merging-technique-level Effects. $p < 0.05$ indicates statistical significance.

| Task Setting | Merging Tech. | Task |
|---|---|---|
| Glue | 0.575 | $2.357 \times 10^{-36}$ |
| Lots-of-LoRAs | 0.987 | $1.699 \times 10^{-7}$ |

We further conduct statistic tests on how merging technique and merged task are correlated to merging loss. The statistical evidence presented in Table 3 provides compelling support for task-level incompatibility as the primary cause of merging collapse. Both tables show a stark contrast in p-values between merging technique-level and task-level effects. These consistent findings across different datasets and model architectures strongly suggest that merging collapse stems primarily from fundamental task incompatibilities rather than limitations in merging techniques. The choice of merging technique appears to have minimal impact on performance compared to the inherent compatibility between task representations, confirming our hypothesis that certain tasks face intrinsic barriers to successful merging regardless of the techniques employed.

> **Answer to RQ2:** Statistical analysis reveals that merging collapse is primarily task-dependent rather than method-dependent. The highly significant task-level effects contrasted with non-significant merging technique-level effects across both datasets demonstrate that inherent task incompatibilities, not methodological limitations, are the key to merging collapse.

## 3.4 RQ3: CORRELATIVE FACTORS FOR MERGING COLLAPSE

### 3.4.1 THEORETICAL EXPLANATION

Our empirical results establish a clear pattern, where merging collapse is not determined methodological choices but inherent task incompatibility.

To shed light on these empirical results with theoretical analysis, we develop a theoretical framework that formalizes the relationship between representational incompatibility and merging collapse.

By modeling the merging process through the lens of rate-distortion theory Berger (2003), we can establish lower bounds on the minimum distortion achievable when combining representations from different tasks. Under the assumption of LMC Frankle et al. (2020), which is widely observed in pre-training and fine-tuning paradigm Zhou et al. (2024), we prove the following theorem.

**Theorem 1** (Hidden–State Diameter Controls Mergeability). *Let $\{\theta_i\}_{i=1}^N \subset \mathbb{R}^p$ be $N$ fine-tuned minima of the same base network $F(\cdot; \theta)$, and let $h(x; \theta) \in \mathbb{R}^d$ be a fixed hidden layer. Assume linear mode connectivity (LMC): every convex combination $\sum_i \alpha_i \theta_i$ ($\alpha_i \geq 0, \sum \alpha_i = 1$) attains the same training loss $\leq \varepsilon$. Denote*

$$d^2(i,j) = \mathbb{E}_X \|h(X;\theta_i) - h(X;\theta_j)\|_2^2, \qquad \Delta = \max_{i,j} d(i,j)$$

*and for any candidate merged model $\hat{\theta}$ define the worst-case hidden-state distortion $\delta_{\max}(\hat{\theta}) = \max_i \mathbb{E}_X \|h(X;\hat{\theta}) - h(X;\theta_i)\|_2^2$.*

*(i) (Achievability) There exists a convex merge $\bar{\theta} = \sum_i \alpha_i \theta_i$ such that $\delta_{\max}(\bar{\theta}) \leq \frac{d}{2(d+1)}\Delta^2$.*

*(ii) (Converse) Every merge satisfies $\delta_{\max}(\hat{\theta}) \geq \frac{1}{4}\Delta^2$; hence $\frac{1}{4}\Delta^2$ is the minimum attainable distortion $D^\star$.*

*(iii) Viewing $I \sim \text{Unif}\{1,\dots,N\}$ and $Y = h(X;\theta_I)$ as the source, the Shannon rate–distortion curve obeys $R(D) = 0$ iff $D \geq D^\star$, and $R(D) \geq \log_2 N$ for $D < D^\star$.*

*Consequently, under LMC the single scalar $\Delta$ completely characterises (a) whether the experts are mergeable within a budget $D_{budget}$ and (b) the zero-rate point of the rate–distortion function.*

*Sketch.* Since LMC implies *linearity of hidden states in parameter space*, the hidden representation of any convex merge lies in the convex hull of $\{h(\cdot; \theta_i)\}$. The minimum enclosing ball of a finite set in $\mathbb{R}^d$ has radius less than $\frac{d}{2(d+1)}$ of its diameter (Jung's Theorem Jung (1901)). Details are in our Appendix C. □

### 3.4.2 HIDDEN-STATE DISTANCE SIMILARITY.

Theorem 1 implies that the representations' distances between different models fine-tuned on different tasks can be an effective metric to capture the task-level merging collapse. Inspired by theorem 1, we propose Hidden-state Distance Similarity which is calculated based on the average L2-distance between the hidden states of different models processing a same set of input, referred as $d_{i,j}$. Assuming we calculate hidden-state distance similarity over a group of models $\{1, 2, ..., n\}$ and their hidden states' L2-distances $D = d_{1,2}, d_{1,3}, ... d_{n-1,n}$, hidden-state distance similarity is calculated through:

$$HiddenSim(i,j) = \begin{cases} 1 & i = j \\ \frac{max(D) - d_{i,j}}{max(D) - min(D)} & i \neq j \end{cases} \tag{2}$$

, where $HiddenSim$ stands for the Hidden-state distance similarity. In our experiments measuring hidden state distances, we draw $k = 5$ datapoints from every task's dataset and compose them into a validation dataset. By running models on the validation dataset, we compare their hidden states of the last layer by calculating the normalized L2 distance averaged across each datapoint.

### 3.4.3 EMPIRICAL RESULTS ON CORRELATION WHEN MERGING PAIRS OF MODELS

To validate our theoretical analysis with empirical results, we first investigate the correlation between conflict metrics and merging collapse.

Table 4: P-values of Pearson correlation coefficient Benesty et al. (2009) on how conflict metrics are correlated to merging loss with different merging techniques when merging two model checkpoints. $p < 0.05$ indicates statistical significance.

| Conflict Metrics | Merging Techniques | | | | |
|---|---|---|---|---|---|
| | TIES | LA | DARE | SLERP | TA |
| Parameter Magnitude Change Ratio | 0.192 | 0.098 | 0.834 | 0.085 | 0.186 |
| Parameter Sign Change Ratio | 0.379 | 0.460 | 0.882 | 0.408 | 0.659 |
| Conflicting Parameter Magnitude Change Ratio | 0.170 | 0.170 | 0.979 | 0.156 | 0.235 |
| Average Cosine Similarity | 0.262 | 0.459 | 0.635 | 0.444 | 0.296 |
| **Hidden State Distance Similarity** | **0.006** | **0.001** | 0.145 | **0.001** | **0.006** |

**Experiment Setting.** As the conflict metrics are defined between two models, we perform additional experiment merging all the pairs ($C_8^2 = 28$ pairs) of Qwen2.5-3B checkpoints on GLUE dataset across five merging techniques. Though merging two models is much easier than merging eight, merging collapse persists in certain task combinations as shown in Appendix Table 18.

**Results.** Contrary to intuitive expectations, the widely adopted parameter update conflicts we introduced in Section 2 exhibit remarkably weak correlations with merging collapse across all four parameter update conflict metrics. This absence of correlation of existing parameter update conflict metrics is particularly evident in Table 4, where we calculate the p-values of Pearson correlation coefficient Benesty et al. (2009) to analyze how existing parameter conflict metrics are correlated to merging loss. With all p-values $> 0.05$, though we can't sufficiently conclude that existing parameter conflict metrics has completely no correlation with merging loss, it is inarguable that they perform poorly to capture the underlying mechanisms driving merging collapse, further indicating that the phenomenon likely emerges from more complex, task-related incompatibilities rather than straightforward parameter update disagreements.

Our hidden state distance similarity metric, on the other hand, shows a strong correlation with merging collapse. Except for DARE, which is found to have unstable merging performance like discussed in RQ1, all p-values of Pearson correlation coefficient are much less than the threshold of $p = 0.05$. Even for DARE, our metric keeps a drastic advantage over existing metrics in correlation. Our metric corresponds significantly better to the actual merging performance than existing parameter update conflicts.

### 3.4.4 Empirical results on HiddenSim for scaled merging.

We further extend our theory explanation to the situation where more models are merged together.

**Experiment Setting.** We perform our empirical investigation on four 8-task groups: one task group consisting of 8 Qwen2.5-3B checkpoints fine-tuned on GLUE tasks, and three task groups (a), (b), (c), each containing of 8 Lots-of-LoRAs tasks whose detail composition can be seen in Appendix Table 23.

**Visualization.** Figure 1 demonstrates a strong correlation between hidden state similarity and merging performance in merging eight Qwen2.5-3B checkpoints in GLUE tasks, where most tasks show high similarity scores with other tasks, while certain tasks like QQP and WNLI exhibit significantly lower scores with other tasks, explaining their poor merging outcomes in a certain way. This pattern extends to LoRA task groups, where we adopt similar experiments on three Lots-of-LoRAs task groups (a), (b), (c) and find out similar correlation between lower similarity score and awful merging performance or even merging collapse, which is presented in our Appendix Figure 3 ~ 5.

**Quantitative Analysis.** To better understand how hidden state similarity score is correlated to merging performance and conduct quantitative analysis, we define the *Merging Difficulty Score (MDS)* for each task $i$ as the reciprocal of average representational similarity scores with other tasks ($j \neq i$):

$$MDS_i = \frac{1}{\text{Average}_{j \neq i}(HiddenSim(i,j))} \tag{3}$$

This metric, inspired by the resistance in physics, provides a non-linear measure of similarity scores amplifying the impact of low similarities where higher MDS indicates worse mergeability (higher merging loss), i.e., greater resistance to merging.

In Table 7~ 10, we present the calculated MDS value and bold the columns with the most severe merging collapse across all merging techniques. Without exception, all tasks with merging collapse have a significantly larger MDS. A statistical analysis is also shown in Table 5, which shows the p-values of Pearson correlation coefficients between every task's MDS and *best* merging loss across different merging techniques. Among all four eight-task groups, all p-values are lower than the threshold $p = 0.05$. Results in these tables support our expectation of MDS that higher MDS indicates worse mergeability, providing further empirical evidence of our theory.

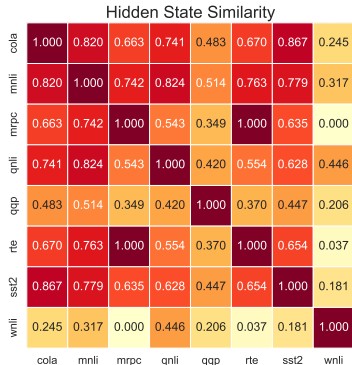

Figure 1: Heatmap of similarity score of hidden states in GLUE tasks.

Table 5: P-values of Pearson correlation coefficients. on how MDS is correlated to the *best* merging loss in merging Qwen2.5-3B checkpoints on GLUE and Lots-of-LoRAs task groups (a), (b) and (c) in experiments. $p < 0.05$ indicates statistical significance.

| Merging tasks | p-value |
|---|---|
| GLUE | **0.0032 ≪ 0.05** |
| group (a) | **0.0013 ≪ 0.05** |
| group (b) | **0.0288 < 0.05** |
| group (c) | **0.0059 ≪ 0.05** |

These findings from both intuitive figures and quantitative analysis confirm that representational compatibility at the hidden state level, rather than parameter update conflicts, is more correlated with merging success or collapse, validating our theoretical analysis that merging collapse stems from differences in model representations.

**Merging Tasks Selection Guiding.** Merging models with higher similarity scores could lead to better merging performance. In group (a), the 6-th task incurs significant merging collapse. Using our MDS metric as a guide, we strategically replace this task with more compatible alternatives from Lots-of-LoRA tasks and construct task gropus (a1) and (a2). Comparing their merging performance with TIES as shown in Table 6, we find that merging tasks with lower MDS do achieve lesser merging collapse, further supporting our theory explanation and demonstrating the potential usefulness of our findings.

Table 6: Comparison among three task groups.

| | (a) | (a1) | (a2) |
|---|---|---|---|
| Worst Loss | -70.8 | -17.8 | -19.1 |
| Worst MDS | 5.05 | 2.52 | 2.89 |

Table 7: Detail data when merging 8 checkpoints from GLUE dataset.

| | | | | | | | | |
|---|---|---|---|---|---|---|---|---|
| LA | -2.1 | -2.4 | -15.6 | -8.2 | -25.7 | -5.6 | -1.8 | **-48.1** |
| TA | -1.9 | -2.5 | -15.6 | -8.1 | -25.7 | -5.2 | -1.9 | **-48.1** |
| TIES | -4.3 | -0.7 | -6.6 | -26.9 | -24.9 | -10.4 | -22.7 | **-30.8** |
| MDS | 1.56 | 1.47 | 1.78 | 1.68 | 2.51 | 1.73 | 1.72 | **4.89** |

Table 8: Detail data when merging group (a) tasks from Lots-of-LoRAs.

| | | | | | | | | |
|---|---|---|---|---|---|---|---|---|
| LA | -9.7 | -6.8 | -0.6 | 17.0 | -4.9 | **-78.0** | -4.6 | -16.5 |
| TA | -17.9 | -18.7 | -3.7 | 12.8 | -6.3 | **-78.6** | -6.4 | -11.4 |
| TIES | -10.8 | -6.0 | -3.7 | -0.9 | -5.6 | **-70.8** | -3.1 | -15.5 |
| MDS | 2.85 | 1.99 | 2.90 | 1.64 | 1.68 | **5.05** | 1.98 | 2.65 |

Table 9: Detail data when merging group (B) tasks from Lots-of-LoRAs.

| | | | | | | | | |
|---|---|---|---|---|---|---|---|---|
| LA | 2.5 | -9.1 | -30.0 | -11.7 | -10.7 | **-75.8** | -40.2 | 0.7 |
| TA | -16.0 | -16.3 | -57.5 | -14.7 | -17.6 | **-78.9** | -44.7 | -9.9 |
| TIES | -4.1 | -6.8 | -39.1 | -9.8 | -14.5 | **-68.2** | -54.8 | -0.9 |
| MDS | 2.22 | 2.78 | 1.62 | 2.19 | 1.45 | **8.15** | 1.94 | 1.43 |

Table 10: Detail data when merging group (C) tasks from Lots-of-LoRAs.

| | | | | | | | | |
|---|---|---|---|---|---|---|---|---|
| LA | 2.9 | -15.2 | **-62.8** | -18.6 | -13.1 | -5.8 | -8.3 | -1.0 |
| TA | -7.3 | -40.7 | **-67.6** | -22.3 | -10.4 | 26.3 | -35.5 | -15.7 |
| TIES | -0.1 | -32.6 | **-57.1** | -5.9 | -14.7 | 6.6 | -4.5 | -1.0 |
| MDS | 1.45 | 1.30 | **3.75** | 1.23 | 1.70 | 1.23 | 1.24 | 1.28 |

**Answer to RQ3:** While parameter-level conflicts show little correlation with merging collapse across multiple conflict calculation methods, hidden state similarity strongly correlates with merging success, with higher representational compatibility corresponding to better merged performance. This confirms that merging collapse stems from fundamental differences in task representations rather than parameter disagreements, explaining why catastrophic degradation exists across different merging techniques.

## 4 RELATED WORK

### 4.1 DEVELOPMENT OF LLMS

LLMs are increasingly being deployed across a wide range of application scenarios Bommasani et al. (2021). These application scenarios span a spectrum of specialized domains, ranging from intelligent customer service systems requiring precise query resolution Xiaoliang et al. (2024); Pandya & Holia (2023) to clinical decision-support tools in healthcare diagnostics Abbasian et al. (2023); Zhang et al. (2024), from real-time risk assessment frameworks in financial operations Zhao et al. (2024); Hasan et al. (2020) to adaptive learning platforms for personalized educational recommendations Urdaneta-Ponte et al. (2021). This substantial diversity in task requirements and operational contexts poses unique challenges when adapting general-purpose LLMs to domain-specific downstream tasks. To enhance targeted capabilities of LLMs in specialized applications, fine-tuning has emerged as a predominant methodological paradigm Yang et al. (2019); Brown (2020); Raffel et al. (2023), enabling the alignment of pre-trained knowledge with task-specific objectives on curated domain datasets. The pre-training and fine-tuning paradigm makes linear mode connectivity Nagarajan & Kolter (2019); Frankle et al. (2020)—where solutions in parameter space can be linearly interpolated while maintaining performance—as an emergent property of fine-tuned models Qin et al. (2022); Zhou et al. (2024).

### 4.2 MODEL MERGING

There is a growing body of work on post-training model merging Wortsman et al. (2022); Zhang et al. (2019); Ilharco et al. (2023); Chitale et al. (2023); Ortiz-Jimenez et al. (2024); Yadav et al. (2024a); Yu et al. (2024), which focuses on combining pre-trained models that have been fine-tuned on different tasks. Linear averaging approaches Wortsman et al. (2022); Zhang et al. (2019) average parameters of multiple fine-tuned models element-wise. Task arithmetic Ilharco et al. (2023); Chitale et al. (2023); Ortiz-Jimenez et al. (2024) leverages task-specific weight vectors, to represent weights for specific tasks to facilitate direct manipulation of multi-task learning by adding or subtracting on task vectors. These approaches are vulnerable to parameter update conflicts Chen & Kwok (2024); Kong et al. (2024). TIES Yadav et al. (2024a) tries to reduce parameter update conflicts with heuristics such as trimming minor updates to avoid redundant updates Zbontar et al. (2021). DARE Yu et al. (2024) improves upon TIES by re-scaling the remaining parameters to keep the distributional stability. Twin-merging Lu et al. (2024) and AdaMerging Yang et al. (2023) relaxes the storage constraints to improve the model performance after merging. Despite these advances, a fundamental understanding of merge conflicts remains underdeveloped.

## 5 CONCLUSION

In this paper, we have investigated the fundamental limitations of model merging and identified the key factors that determine merging success or failure. Our research reveals that task-level representational incompatibility is the primary driver of merging collapse, with certain task combinations consistently failing across all merging methods. This finding challenges the conventional wisdom that parameter conflicts are the main obstacle to successful merging. We further develop a theoretical framework based on rate-distortion theory that establishes relationship between model representation distances and merging collapse, verified by empirical evidence.

## 6 ETHICS STATEMENT

This work adheres to the ICLR Code of Ethics. No human subjects or animal experimentation were involved. All datasets used were obtained in compliance with relevant usage guidelines, ensuring no privacy violations. We have carefully avoided biases or discriminatory outcomes. No personally identifiable information was used, and no experiments posed privacy or security risks. We are committed to transparency and integrity throughout the research process.

## 7 REPRODUCIBILITY STATEMENT

We have taken steps to ensure that the results presented in this paper are reproducible. All code and datasets are publicly available or provided in the supplementary material to facilitate replication. The experimental setup is described in detail.

Furthermore, the public datasets used are openly accessible, ensuring consistent and reproducible evaluation. These measures aim to enable other researchers to replicate our work and advance the field.

## 8 LLM USAGE

Large Language Models (LLMs) were employed solely to assist in writing and polishing the manuscript, including refining language, improving readability, and enhancing clarity. The LLM was used for tasks such as sentence rephrasing, grammar checking, and improving overall flow.

The LLM was not involved in ideation, research methodology, or experimental design. All research concepts, analyses, and results were developed and conducted by the authors. The authors take full responsibility for the manuscript content, including any text generated or polished by the LLM, and confirm that all LLM-assisted text adheres to ethical guidelines and does not constitute plagiarism or scientific misconduct.

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

APPENDIX

## A  DETAILED PRELIMINARY

LLMs are large-scale neural networks Bommasani et al. (2021) pre-trained on vast amounts of data that serve as versatile base models for various downstream applications. The development lifecycle of LLMs typically follows a two-stage paradigm: pre-training and fine-tuning.

**Pre-training.** Pre-training serves as the foundational stage in which models are exposed to large and diverse datasets through self-supervised learning, allowing them to acquire broad, general-purpose representations of language and other modalities. As a result, the pre-trained model develops a wide-ranging understanding that makes it adaptable to a variety of downstream tasks. However, the representations learned during pre-training are not optimized for any specific application, necessitating further adaptation to meet the requirements of particular tasks or domains.

**Fine-tuning.** Fine-tuning involves further training a pre-trained model on task-specific data with targeted objectives. Let $M_\theta$ denote a Foundation Model (FM) with trainable parameters $\theta$, which can represent either the entire model for full fine-tuning or a subset of parameters for parameter-efficient fine-tuning techniques Hu et al. (2022); Liu et al. (2022); Houlsby et al. (2019). In collaborative FM development environments, multiple teams often independently fine-tune the same base model for different capabilities or on different data distributions Touvron et al. (2023), resulting in a collection of specialized models. Consider a set of tasks $\mathcal{T} = \{t_1, t_2, ..., t_n\}$. The based FM $M_{\theta_0}$ is fine-tuned on each task $t_i$, resulting in fine-tuned model $M_{\theta_i}$ with updated parameters $\theta_i$. The distributed development creates the need for effective approaches to consolidate these independently evolved models.

**Model merging.** Building upon the independent fine-tuning efforts described above, there arises a crucial need to integrate the strengths of multiple specialized models. Model merging Wortsman et al. (2022); Zhang et al. (2019); Ilharco et al. (2023); Chitale et al. (2023); Ortiz-Jimenez et al. (2024); Yadav et al. (2024a); Yu et al. (2024); Yang et al. (2024b) refers to the process of combining multiple independently fine-tuned models into a unified model that preserves the capabilities of its constituents. Given the updated parameters $\{\theta_1, ..., \theta_n\}$ obtained from fine-tuning the common base model $M_{\theta_0}$, the goal of model merging is to produce a consolidated model $M_{\theta_{merged}}$ that preserves the task-specific knowledge from each fine-tuned model.

**Task vectors Hendel et al. (2023); Hojel et al. (2024); Ilharco et al. (2023); Chitale et al. (2023); Ortiz-Jimenez et al. (2024).** A task vector provides a concise representation of the model update resulting from fine-tuning on a specific task. Formally, let $\theta_0 \in \mathbb{R}^d$ denote the pre-trained model weights, and $\theta_t \in \mathbb{R}^d$ the weights after fine-tuning on task $t$. The task vector is defined as $\tau_t = \theta_t - \theta_0$, capturing both the direction and magnitude of parameter changes induced by fine-tuning. In other words, $\tau_t$ encodes the element-wise modifications made to the model parameters for adapting to task $t$. Furthermore, task vectors enable a straightforward approach to model merging: by averaging the task vectors from $n$ tasks, one can construct a merged model as $\theta_{merged} = \theta_0 + (\tau_1 + \tau_2 + \cdots + \tau_n)/n$.

**Parameter update conflicts.** When merging models fine-tuned on different tasks, conflicts often arise within the parameter updates Yadav et al. (2024a); Yang et al. (2024b). A direct and intuitive form of conflict occurs when the elements in task vectors for different tasks have opposite signs, implying that the optimal parameter update for one task may be detrimental to another Yadav et al. (2024a). However, even in the absence of sign differences, significant disparities in the magnitudes of updates can also lead to suboptimal performance in the merged model.

**Parameter update conflict metrics.** Based on established literature Huang et al. (2024); Yadav et al. (2024a); Ilharco et al. (2023), we focus on these four metrics to provide a comprehensive view of parameter update conflicts.

- **Parameter Magnitude Change Ratio** is the ratio between the sum of the difference of each parameter of the same position and the sum of two models' task vectors' magnitude:

$$\text{Parameter Magnitude Change Ratio}(i, j) = \frac{\|\theta_i - \theta_j\|_1}{\|\tau_i\|_1 + \|\tau_j\|_1} \qquad (4)$$

- **Parameter Sign Change Ratio** is the ratio between the number of positions where two models' task vector not sharing a same sign and the total number of parameters positions.

$$\text{Parameter Sign Change Ratio}(i,j) = \frac{\#[sign(\tau_i) \neq sign(\tau_j)]}{\#(parameter in \tau_i)} \tag{5}$$

- **Conflicting Parameter Magnitude Change Ratio** is the ratio between the difference of parameters at positions where two models' task vector not sharing a same sign and the sum of two models' task vectors' magnitude.

$$\text{Conflicting Parameter Magnitude Change Ratio}(i,j) = \frac{\|(\theta_i - \theta_j)[sign(\tau_i) \neq sign(\tau_j)]\|_1}{\|\tau_i\|_1 + \|\tau_j\|_1} \tag{6}$$

- **Average cosine Similarity Between Pair's Task Vectors** stands for the average cosine similarity between two model's parameter vectors.

$$\text{Average cosine Similarity Between Pair's Task Vectors}(i,j) = Average_{\tau_i, \tau_j \in \theta_i, \theta_j} \frac{\tau_i \cdot \tau_j}{\|\tau_i\|_2 \|\tau_j\|_2} \tag{7}$$

**Merging collapse**. Similar to model collapse training on synthesis data Shumailov et al. (2024), we introduce the concept of merging collapse as phenomenon when distinct fine-tuned models cannot be successfully combined by a given model merging technique while preserving their original capabilities. Formally, let $P(\theta_i, T_i)$ denote the performance of model $M_{\theta_i}$ on task $T_i$, we quantify merging collapse using the **merging loss** $-100\% \leq L(T_i) \leq 0\%$ on each task $T_i$:

$$L(T_i) = \left(\frac{P(\theta_{\text{merged}}, T_i)}{P(\theta_i, T_i)} - 1\right) \times 100\% \tag{8}$$

As for multiple tasks, we further define the average merging loss across all tasks as the arithmetic mean. In the analysis that follows, we omit the percentage sign "%" for simplicity.

**ANOVA F-test Maxwell et al. (2017) and Pearson Correlation Coefficient Benesty et al. (2009)**. In our statistical analysis, we adopt many tests by calculating ANOVA F-test and Pearson Correlation Coefficient's p-value. ANOVA F-tests are statistical methods used to determine whether there are statistically significant differences between the means of independent groups. Here we use ANOVA F-tests to identify which factors are statistically significant to the variations in data. The ANOVA F-test works by comparing the amount of variance between the groups to the amount of variance within the groups. Pearson correlation coefficient measures the strength and direction of a linear relationship between two continuous variables. Here we use the p-value of it to estimate whether the relationship exists or not.

## B    ELABORATION ON CLOSELY RELATED WORK

The field of model merging has seen significant development in recent years, with a predominant focus on parameter update conflicts as the central challenge when combining multiple fine-tuned models Yadav et al. (2024a); Yu et al. (2024); Yadav et al. (2024b). This perspective has driven researchers to propose increasingly sophisticated techniques Yadav et al. (2024a); Yu et al. (2024); Lu et al. (2024) aimed at alleviating these conflicts through various parameter manipulation strategies. Below, we provide a comprehensive analysis of these methods, examining their theoretical foundations, technical implementations, underlying assumptions, and limitations.

- **Linear Averaging (LA)** Zhang et al. (2019); Wortsman et al. (2022) represents one of the earliest and most straightforward approaches to model merging, where parameters of multiple fine-tuned models are averaged element-wise. Formally, for $n$ models with parameters $\theta_1, \theta_2 ..., \theta_n$ , the merged model parameters are computed as $\theta_{merged} = \frac{1}{n} \sum_{i=1}^{n} \theta_i$.

  This approach gained popularity due to its simplicity and surprising effectiveness across certain tasks. The theoretical justification often cited is that averaging can be viewed as an ensemble method in parameter space rather than in prediction space, potentially reducing

variance while preserving useful features learned by individual models. LA makes the implicit assumption that parameter conflicts are random or normally distributed, such that averaging will converge toward "true" parameter values while canceling out noise.

However, LA treats all parameters equally without considering their relative importance or the nature of the tasks being merged. This naive averaging can lead to sub-optimal performance when combining models fine-tuned on substantially different tasks, as conflicting parameter updates may nullify each other rather than complement one another. Extensions of LA have explored weighted averaging schemes that assign different importance to different models, but the core limitation persists: LA addresses parameter conflicts through simple arithmetic operations without considering the semantic compatibility of the underlying tasks.

- **Task Arithmetic (TA)** Ilharco et al. (2023); Chitale et al. (2023); Ortiz-Jimenez et al. (2024) represents a more sophisticated approach that interprets model fine-tuning as vector operations in parameter space. The core insight is that the difference between a fine-tuned model and its pre-trained base represents a "task vector" that encapsulates the knowledge specific to that task.

  Mathematically, for a pre-trained model with parameters $\theta_{pre}$ and a fine-tuned model with parameters $\theta_{ft}$, the task vector is defined as $\vec{t} = \theta_{ft} - \theta_{pre}$. These task vectors can then be manipulated through addition, subtraction, and scaling operations to compose new capabilities or transfer knowledge between models. For instance, to merge $n$ tasks, one might compute $\theta_{merged} = \theta_{pre} + \sum_{i=1}^{n} \alpha_i \vec{t_i}$, where $\alpha_i$ represents a scaling factor for each task vector.

  TA has been particularly successful in transferring specific capabilities between models and enabling "editing" of model behaviors. The framework has been extended to various settings, including LoRA-based fine-tuning Chitale et al. (2023) and tangent space operations Ortiz-Jimenez et al. (2024), which interpret task vectors as directions in a manifold of model capabilities rather than as simple Euclidean vectors.

  The approach implicitly assumes that parameter updates for different tasks operate in independent or at least non-interfering subspaces of the overall parameter space. When this assumption holds, task vectors can be meaningfully combined without destructive interference. However, this assumption breaks down when tasks require conflicting parameter updates in the same regions of the network, leading to potential performance degradation. TA also typically requires careful tuning of scaling factors $\alpha_i$ to balance the influence of different task vectors, adding complexity to the merging process.

- **TIES (Trim, Elect, and Integrate for Sign-conflict)** Yadav et al. (2024a) directly addresses the parameter conflict problem by proposing a three-stage approach to reconcile conflicting updates. The method explicitly identifies and resolves sign conflicts that occur when merging multiple fine-tuned models.

  In the Trimming stage, TIES identifies parameters with small update magnitudes across all models and resets them to their pre-trained values, based on the assumption that small updates represent noise rather than task-specific learning. Formally, for each parameter position $j$, if $|\theta_{i,j} - \theta_{pre,j}| < \epsilon$ for all models $i$, then $\theta_{merged,j} = \theta_{pre,j}$.

  The Election stage handles sign conflicts by identifying parameters where different models disagree on the direction of update (positive or negative). TIES employs a voting mechanism where the sign with the largest cumulative magnitude across models wins. Specifically, for each conflicting parameter position $j$, TIES computes $S_+ = \sum_{i:\Delta\theta_{i,j}>0} |\Delta\theta_{i,j}|$ and $S_- = \sum_{i:\Delta\theta_{i,j}<0} |\Delta\theta_{i,j}|$, where $\Delta\theta_{i,j} = \theta_{i,j} - \theta_{pre,j}$. If $S_+ > S_-$, a positive update is chosen; otherwise, a negative update is selected.

  Finally, the Integration stage combines the updates using a weighted average scheme based on the update magnitudes. This approach gives more influence to models with larger parameter changes, reflecting the assumption that larger updates represent more decisive learning for specific tasks.

  TIES represents one of the most direct attempts to address parameter conflicts, explicitly modeling the merging problem as a conflict resolution challenge. The method has shown improved performance over simple averaging, particularly when merging models with divergent fine-tuning objectives. However, TIES still operates entirely at the parameter level, without considering the semantic relationships between tasks or the representational

compatibility in the model's hidden space. Its effectiveness is predicated on the assumption that parameter conflicts are the primary obstacle to successful merging.

- **DARE (Drop And REscale)** Yu et al. (2024) introduces a sparsification approach to model merging that selectively preserves only the most significant parameter updates from each fine-tuned model. The core insight of DARE is that many parameter updates during fine-tuning may be redundant or noisy, and by focusing on the most important changes, merging can be more effective.

  The DARE procedure consists of two main steps: dropping and rescaling. In the dropping phase, for each fine-tuned model $i$, DARE computes the update vector $\Delta\theta_i = \theta_i - \theta_{pre}$ and retains only the top-k% updates based on absolute magnitude. This creates a sparse update vector $\Delta\theta_i^{sparse}$ where most elements are zero.

  In the rescaling phase, DARE adjusts the magnitude of the remaining updates to compensate for the dropped values, ensuring that the overall "energy" of the update is preserved. The rescaling factor is typically computed as $\gamma = \frac{\|\Delta\theta_i\|_2}{\|\Delta\theta_i^{sparse}\|_2}$, and the final sparse update becomes $\Delta\theta_i^{final} = \gamma \cdot \Delta\theta_i^{sparse}$.

  The merged model is then created by combining these sparse updates, often using a weighted sum approach: $\theta_{merged} = \theta_{pre} + \sum_{i=1}^{n} w_i \cdot \Delta\theta_i^{final}$, where $w_i$ are weights assigned to each model.

  DARE's sparsification strategy effectively reduces the potential for parameter conflicts by eliminating most of the parameter updates, focusing only on those deemed most important for each task. This approach has shown strong results, particularly when merging models fine-tuned on diverse tasks. However, DARE requires careful tuning of the sparsity level (k%), and like other parameter-focused methods, it does not directly consider the semantic compatibility between tasks. The method also implicitly assumes that the magnitude of parameter updates correlates with their importance, which may not always hold true, especially in complex neural networks where small but critical updates can significantly influence model behavior.

- **SLERP (Spherical Linear Interpolation)** Freeden & Törnig (1981); Goddard et al. (2024) represents a geometrically motivated approach to model merging that treats parameter vectors as points on a hypersphere rather than in Euclidean space. Originally developed for computer graphics applications to smoothly interpolate between orientations Freeden & Törnig (1981), SLERP has been adapted to neural network merging to better preserve the directional information encoded in weight vectors.

  The key insight behind SLERP is that neural network parameters often operate through their directional influence rather than their absolute magnitudes. When parameters are viewed as directions on a unit hypersphere, SLERP provides a way to interpolate along the shortest great-circle arc connecting two points on this sphere.

  Mathematically, given two normalized parameter vectors $\hat{\theta}_1 = \frac{\theta_1}{\|\theta_1\|}$ and $\hat{\theta}_2 = \frac{\theta_2}{\|\theta_2\|}$, and an interpolation parameter $t \in [0, 1]$, SLERP computes:

  $$\text{SLERP}(\hat{\theta}_1, \hat{\theta}_2, t) = \frac{\sin((1-t)\omega)}{\sin(\omega)}\hat{\theta}_1 + \frac{\sin(t\omega)}{\sin(\omega)}\hat{\theta}_2$$

  where $\omega = \arccos(\hat{\theta}_1 \cdot \hat{\theta}_2)$ is the angle between the two vectors. For merging multiple models, pairwise SLERP operations can be chained, or generalized versions of spherical interpolation for multiple points can be employed.

  When applied to model merging, SLERP has shown advantages over linear interpolation, particularly when merging models with significantly different parameter magnitudes or when directional information is more important than absolute values. Goddard et al. Goddard et al. (2024) demonstrated SLERP's effectiveness in their ARCEES framework, showing improved performance when merging language models across diverse tasks.

  SLERP's geometric perspective offers a more principled way to combine parameters than simple averaging, but it still operates entirely in parameter space. While it addresses some limitations of linear averaging by better preserving directional information, SLERP does not explicitly consider task compatibility or the semantic relationships between different fine-tuning objectives. The approach also adds computational complexity, as normalization and spherical operations are more expensive than simple linear operations.

- **Fisher Merging** Matena & Raffel (2022) represents an information-theoretic approach to model merging that leverages the Fisher Information Matrix (FIM) to weight parameter updates according to their importance. The core insight is that not all parameters contribute equally to a model's performance on a given task, and the FIM provides a principled way to quantify this importance.

  For each fine-tuned model, the Fisher Information Matrix approximates the second derivative of the loss function with respect to the model parameters, indicating how sensitive the model's outputs are to changes in each parameter. Parameters with high Fisher information are considered more critical for the task.

  Mathematically, Fisher Merging computes the merged parameters as:

  $\theta_{merged} = \theta_{pre} + \sum_{i=1}^{n} F_i^{-1} \cdot (\theta_i - \theta_{pre})$

  where $F_i$ is the Fisher Information Matrix for model $i$. In practice, computing the full FIM is prohibitively expensive for large models, so various approximations are used, such as diagonal approximations that only consider the diagonal elements of the matrix.

  Fisher Merging has shown strong performance in scenarios where tasks have partially overlapping parameter importance, as it effectively allows each task to dominate in the parameter regions most critical to its performance. However, the approach requires estimating Fisher information for each fine-tuned model, adding computational overhead. Moreover, the method still operates under the assumption that parameter conflicts are the primary challenge in model merging, albeit with a more sophisticated weighting scheme than simple averaging.

- **RegMean** Jin et al. (2022) approaches model merging from a regularization perspective, formulating the merging problem as finding a compromise between staying close to the average of fine-tuned models while minimizing the expected loss across tasks. The method does not require access to the original training data, making it practical for many real-world scenarios.

  RegMean introduces a regularization term that penalizes the merged model for deviating from the simple average of fine-tuned models. The optimization objective can be written as:

  $\theta_{merged} = \arg\min_{\theta} \sum_{i=1}^{n} \mathcal{L}_i(\theta) + \lambda \|\theta - \frac{1}{n} \sum_{i=1}^{n} \theta_i\|^2$

  where $\mathcal{L}_i$ is the loss function for task $i$ and $\lambda$ is a regularization parameter. This objective is typically optimized using gradient-based methods, starting from the average of fine-tuned models.

  By balancing task-specific performance with proximity to the average model, RegMean can discover parameter configurations that resolve conflicts more effectively than simple averaging. The approach has shown particular strength when merging models fine-tuned on tasks with competing objectives.

  However, RegMean requires computing gradients for each task during the merging process, which can be computationally intensive for large models or many tasks. Like other methods, it still fundamentally operates on the assumption that finding the right compromise in parameter space is the key to successful merging.

In contrast to these parameter-centric approaches, our work fundamentally challenges the prevailing assumption that parameter-level conflicts are the primary determinant of merging success or failure. Through extensive empirical analysis across diverse merging scenarios, we demonstrate that conventional metrics of parameter conflict show surprisingly minimal correlation with actual merging outcomes. Models with substantial parameter disagreements can merge successfully, while others with seemingly minor conflicts experience catastrophic collapse.

Instead, our research reveals that hidden state similarity serves as a remarkably strong predictor of merging outcomes. This metric, which measures the compatibility of internal representations between models, consistently correlates with successful merging across both GLUE benchmark tasks and specialized LoRA fine-tuning scenarios. Our findings indicate that when two models encode similar representational structures in their hidden states—even if their parameter values differ significantly—they can be successfully merged. Conversely, models with fundamentally different representational structures will collapse when merged, regardless of parameter-level agreement.

This insight represents a paradigm shift in understanding model merging: the success of merging depends not on numerical alignment of parameters, but on the compatibility of the underlying task

representations in the model's internal feature space. This explains several previously puzzling observations in model merging literature, such as why some seemingly similar tasks fail to merge while others succeed unexpectedly. Rather than focusing exclusively on reconciling parameter conflicts, the field should consider methods that explicitly optimize for representational compatibility or that operate directly in the hidden state space. Our findings suggest that successful merging strategies should prioritize preserving the integrity of internal representations over minimizing parameter disagreements, potentially opening new avenues for more effective model combination techniques.

## C PROOF OF THEOREM 1

**Lemma 1** (Jung's Theorem Jung (1901)). *For any finite set $\mathcal{S} \subset \mathbb{R}^d$ with diameter* $\mathrm{diam}(\mathcal{S}) = \max_{u,v \in \mathcal{S}} \|u - v\|_2 = \Delta$, *there exists a point $c \in \mathrm{conv}(\mathcal{S})$ such that*

$$\max_{s \in \mathcal{S}} \|c - s\|_2 \ \le \ \sqrt{\frac{d}{2(d+1)}} \, \Delta.$$

*Proof.* The centre of the minimum enclosing ball of $\mathcal{S}$ has the stated properties and lies in the convex hull (by classic results on the smallest enclosing ball in Euclidean space Megiddo (1983)). $\square$

*Proof of Theorem 1.* Throughout we abbreviate $H_i(x) = h(x; \theta_i)$ for the unique input $x$.

**Step 1: Achievability** ($D^\star \le \frac{d}{2(d+1)} \Delta^2$). Fix any coefficients $\alpha = \{\alpha_i\}_{i=1}^N$ with $\alpha_i \ge 0$, $\sum_i \alpha_i = 1$. LMC ensures that the hidden state of their convex merge satisfies

$$h(x; \bar{\theta}) = h\Big(x; \sum_i \alpha_i \theta_i\Big) = \sum_i \alpha_i \, h(x; \theta_i) = \sum_i \alpha_i H_i(x) \ \in \mathrm{conv}\{H_i(x)\}_i. \tag{9}$$

Apply Lemma 1 to the finite set $\{H_i(x)\}_i$: choosing *the same* convex coefficients $\alpha$ ensures that $h(x; \bar{\theta})$ coincides with the centre $c(x)$ of the minimum enclosing ball, hence $\|h(x; \bar{\theta}) - H_i(x)\|_2 \le \sqrt{\frac{d}{2(d+1)}} \, \Delta_x$ for every $i$, where $\Delta_x = \max_{j,k} \|H_j(x) - H_k(x)\|_2 \le \Delta$. Taking expectations and the maximum over $i$ yields $\delta_{\max}(\bar{\theta}) \le \frac{d}{2(d+1)} \Delta^2$.

**Step 2: Converse** ($D^\star \ge \frac{1}{4} \Delta^2$). Let $\hat{\theta}$ be any merged model and choose indices $(i, j)$ that realise the diameter $d(i, j) = \Delta$. The triangle inequality gives

$$d(i, j) \ = \ \sqrt{\mathbb{E}_X \|H_i(X) - H_j(X)\|_2^2} \ \le \ \sqrt{\delta_i(\hat{\theta})} + \sqrt{\delta_j(\hat{\theta})} \ \le \ 2\sqrt{\delta_{\max}(\hat{\theta})}.$$

Squaring both sides implies $\delta_{\max}(\hat{\theta}) \ge \frac{1}{4} \Delta^2$, proving the lower bound.

**Step 3: Rate–distortion statement.** Let the random pair $(X, I)$ be distributed according to $P_X \times \mathrm{Unif}\{1, \ldots, N\}$, and define $Y = h(X; \theta_I)$. A *zero-rate* encoder cannot send any information about $I$, so the decoder must output a reconstruction $\hat{Y}$ that is independent of $I$ and is therefore produced by a single parameter vector $\hat{\theta}$. The minimum achievable mean-squared error is precisely $D^\star$. Shannon's converse then asserts that $R(D) = 0$ iff $D \ge D^\star$, while for $D < D^\star$ at least $\log_2 N$ bits are necessary to transmit the expert identity, establishing the claimed $R(D)$ behaviour.

**Step 4: Putting the pieces together.** Steps 1 and 2 show $\frac{1}{4} \Delta^2 \le D^\star \le \frac{d}{2(d+1)} \Delta^2$ and identify any convex merge (including the uniform average) as optimal. Step 3 connects this constant to the fundamental rate–distortion limit, completing the proof. $\square$

**Corollary 1** (Practical mergeability test). *Given a distortion budget $D_{budget}$, a set of models is mergeable under LMC iff their hidden–state diameter $\Delta$ satisfies $\frac{d}{2(d+1)} \Delta^2 \le D_{budget}$. Since the hidden dimension $d$ is typically large in language models, $\frac{d}{2(d+1)}$ can be approximated by $\frac{1}{2}$.*

Table 11: Foundation Models Used for Evaluation.

| Model | Training | # of Params | Architecture |
|---|---|---|---|
| **Llama3.2-3B Grattafiori et al. (2024)** | Instruction-tuning | 3B | Decoder-only |
| **Llama3.1-8B Grattafiori et al. (2024)** | Instruction-tuning | 8B | Decoder-only |
| **Qwen2.5-3B Yang et al. (2024a)** | Instruction-tuning | 3B | Decoder-only |
| **Qwen2.5-7B Yang et al. (2024a)** | Instruction-tuning | 7B | Decoder-only |
| **Qwen2.5-14B Yang et al. (2024a)** | Instruction-tuning | 14B | Decoder-only |
| **T5-Base Raffel et al. (2023)** | Task-specific | 220M | Encoder-Decoder |
| **T5-Large Raffel et al. (2023)** | Task-specific | 770M | Encoder-Decoder |
| **T5-XL Raffel et al. (2023)** | Task-specific | 3B | Encoder-Decoder |

Table 12: GLUE Datasets Used in Our Empirical Study.

| Category | Dataset | Type | Domain |
|---|---|---|---|
| | COLA Wang et al. (2018) | CLS. | Grammar Acceptability |
| | MNLI Wang et al. (2018) | CLS. | Textual entailment |
| | MRPC Wang et al. (2018) | CLS. | Semantical equivalence |
| NLP | QNLI Wang et al. (2018) | CLS. | QA Correspondence |
| | QQP Wang et al. (2018) | CLS. | Semantical equivalence |
| | RTE Wang et al. (2018) | CLS. | Textual entailment |
| | SST-2 Wang et al. (2018) | CLS. | Sentiment Prediction |
| | WNLI Wang et al. (2018) | CLS. | Reading comprehension |

## D  EXPERIMENT SETTINGS

**Fine-tuning settings.** All fine-tuning are conducted on an AI server with two Intel Xeon Platinum 8374C 36-core processors, two NVIDIA A100 Graphics Cards with 80G memory, Cuda 12.2, and NVLink Foley & Danskin (2017) enabled for acceleration. When fine-tuning of T5-Base, T5-Large, and T5-XL we use Adam optimizer Kingma (2014) for 1 to 20 epochs on different tasks varied on the size of their dataset with a learning rate of 3e-4, following previous work Yadav et al. (2024a). We fine-tune models with a batch size of 256 with gradient accumulation Paszke et al. (2017) to fit our limited GPU resources. During the fine-tuning process, we use bf16, i.e., brain floating point Burgess et al. (2019), to accelerate the fine-tuning Henry et al. (2019). When fine-tuning the models from the Qwen-2.5 series and the Llama series, we set the learning rate be 1e-5 following previous work Yang et al. (2024a). The Adam optimizer is used with a batch size of 16 with gradient accumulation and trained for 2 to 32 epochs across various tasks to fit the GPU resource limits.

**Merging settings.** When merging models, we set $w_i = 1$ for linear averaging, $\lambda = 0.4$ for task arithmetic, $\lambda = 0.2$ for TIES, $\lambda = 0.1$ for DARE. The hyperparameter values are all the default values recommended by the original paper Yadav et al. (2024a); Ilharco et al. (2023); Yu et al. (2024) or general settings under most circumstances. We have also adopted a small-sized parameter tuning in Appendix F, through which we find out that the model merging collapse persists across different merging hyperparameter values.

## E  DETAILED EXPERIMENTAL RESULTS

### E.1  DETAILED EXPERIMENT RESULTS OF PAIR-MERGING

In Table 1, we show a summary of results of merging 8 fine-tuned checkpoints on GLUE dataset across 8 models of different size. Besides Qwen2.5 models which we have shown the detailed results in Table 2, we show the remaining detailed results in Table 13, Table 14, Table 15, Table 16 and Table 17 here in Appendix.

Table 13: Detail results of merging results among 8 fine-tuned Llama3.2-3B checkpoints in GLUE dataset.

| Method | COLA | MNLI | MRPC | QNLI | QQP | RTE | SST2 | WNLI |
|---|---|---|---|---|---|---|---|---|
| fine-tuned | 80.63 | 87.65 | 87.0 | 92.71 | 86.94 | 90.25 | 95.98 | 54.92 |
| LA | 67.97(-15.69%) | 70.42(-19.65%) | 74.01(-14.92%) | 70.82(-23.61%) | 62.88(-27.67%) | 80.5(-10.79%) | 93.46(-2.62%) | 42.25(-23.07%) |
| TA | 67.68(-16.05%) | 70.41(-19.66%) | 74.01(-14.92%) | 70.93(-23.49%) | 62.87(-27.68%) | 80.14(-11.2%) | 93.57(-2.5%) | 40.84(-25.64%) |
| TIES | 48.22(-40.19%) | 82.39(-5.99%) | 80.88(-7.04%) | 50.72(-45.29%) | 67.26(-22.63%) | 80.14(-11.2%) | 95.29(-0.71%) | 32.39(-41.02%) |
| DARE | 0.0(-100.0%) | 0.0(-100.0%) | 31.37(-63.94%) | 50.53(-45.48%) | 63.02(-27.51%) | 45.48(-49.59%) | 51.6(-46.23%) | 42.25(-23.07%) |
| SLERP | 67.88(-15.81%) | 71.47(-18.45%) | 74.01(-14.92%) | 69.48(-25.05%) | 63.0(-27.52%) | 80.86(-10.4%) | 93.69(-2.38%) | 40.84(-25.64%) |

Table 14: Detail results of merging results among 8 fine-tuned Llama3.1-8B checkpoints in GLUE dataset.

| Method | COLA | MNLI | MRPC | QNLI | QQP | RTE | SST2 | WNLI |
|---|---|---|---|---|---|---|---|---|
| fine-tuned | 84.56 | 90.85 | 89.46 | 95.42 | 87.78 | 92.05 | 97.01 | 73.23 |
| LA | 78.04(-7.7%) | 81.48(-10.31%) | 74.5(-16.71%) | 66.46(-30.34%) | 63.18(-28.02%) | 86.28(-6.27%) | 95.52(-1.53%) | 18.3(-75.0%) |
| TA | 77.94(-7.82%) | 81.48(-10.31%) | 74.75(-16.43%) | 66.5(-30.3%) | 63.17(-28.02%) | 86.28(-6.27%) | 95.41(-1.65%) | 18.3(-75.0%) |
| TIES | 46.97(-44.44%) | 66.72(-26.56%) | 63.97(-28.49%) | 52.29(-45.19%) | 63.18(-28.02%) | 84.47(-8.23%) | 93.8(-3.3%) | 22.53(-69.23%) |
| DARE | 0.0(-100.0%) | 0.0(-100.0%) | 31.61(-64.65%) | 50.53(-47.03%) | 63.18(-28.02%) | 47.29(-48.62%) | 0.0(-100.0%) | 43.66(-40.38%) |
| SLERP | 77.46(-8.39%) | 82.27(-9.44%) | 75.24(-15.89%) | 65.0(-31.88%) | 63.18(-28.02%) | 87.0(-5.49%) | 95.41(-1.65%) | 16.9(-76.92%) |

Table 15: Detail results of merging results among 8 fine-tuned T5-base checkpoints in GLUE dataset.

| Method | COLA | MNLI | MRPC | QNLI | QQP | RTE | SST2 | WNLI |
|---|---|---|---|---|---|---|---|---|
| fine-tuned | 69.12 | 59.77 | 83.57 | 90.68 | 89.63 | 50.54 | 92.54 | 56.33 |
| LA | 69.12(0.0%) | 33.97(-43.15%) | 68.38(-18.18%) | 55.28(-39.03%) | 36.81(-58.92%) | 47.29(-6.42%) | 49.42(-46.59%) | 45.07(-20.0%) |
| TA | 69.12(0.0%) | 31.74(-46.88%) | 68.38(-18.18%) | 51.43(-43.27%) | 36.81(-58.92%) | 46.93(-7.14%) | 49.08(-46.96%) | 47.88(-15.0%) |
| TIES | 30.87(-55.33%) | 34.66(-42.01%) | 68.38(-18.18%) | 51.16(-43.58%) | 36.81(-58.92%) | 48.37(-4.28%) | 49.08(-46.96%) | 57.74(+2.49%) |
| DARE | 30.87(-55.33%) | 32.73(-45.23%) | 68.38(-18.18%) | 41.13(-54.64%) | 36.81(-58.92%) | 35.74(-29.28%) | 49.08(-46.96%) | 29.57(-47.5%) |
| SLERP | 69.12(0.0%) | 31.81(-46.77%) | 68.38(-18.18%) | 50.53(-44.26%) | 36.81(-58.92%) | 47.29(-6.42%) | 49.08(-46.96%) | 56.33(0.0%) |

Table 16: Detail results of merging results among 8 fine-tuned T5-large checkpoints in GLUE dataset.

| Method | COLA | MNLI | MRPC | QNLI | QQP | RTE | SST2 | WNLI |
|---|---|---|---|---|---|---|---|---|
| fine-tuned | 69.12 | 89.24 | 68.38 | 93.84 | 92.16 | 50.54 | 95.41 | 56.33 |
| LA | 68.64(-0.69%) | 36.49(-59.1%) | 68.38(0.0%) | 49.71(-47.02%) | 36.81(-60.05%) | 52.7(+4.28%) | 71.33(-25.24%) | 43.66(-22.5%) |
| TA | 31.16(-54.92%) | 31.84(-64.31%) | 68.38(0.0%) | 50.1(-46.61%) | 36.81(-60.05%) | 48.73(-3.57%) | 49.08(-48.55%) | 57.74(+2.49%) |
| TIES | 49.18(-28.84%) | 39.08(-56.2%) | 68.38(0.0%) | 51.94(-44.64%) | 36.81(-60.05%) | 53.06(+5.0%) | 88.53(-7.21%) | 43.66(-22.5%) |
| DARE | 69.12(0.0%) | 31.81(-64.34%) | 68.38(0.0%) | 50.53(-46.14%) | 36.81(-60.05%) | 47.29(-6.42%) | 49.08(-48.55%) | 56.33(0.0%) |
| SLERP | 69.12(0.0%) | 31.81(-64.34%) | 68.38(0.0%) | 50.53(-46.14%) | 36.81(-60.05%) | 47.29(-6.42%) | 49.08(-48.55%) | 56.33(0.0%) |

Table 17: Detail results of merging results among 8 fine-tuned T5-XL checkpoints in GLUE dataset.

| Method | COLA | MNLI | MRPC | QNLI | QQP | RTE | SST2 | WNLI |
|---|---|---|---|---|---|---|---|---|
| fine-tuned | 81.87 | 31.81 | 72.3 | 79.79 | 63.18 | 56.67 | 94.26 | 53.52 |
| LA | 69.12(-15.57%) | 35.44(+11.39%) | 68.38(-5.42%) | 49.46(-38.01%) | 36.81(-41.73%) | 52.7(-7.0%) | 51.37(-45.49%) | 43.66(-18.42%) |
| TA | 69.12(-15.57%) | 31.91(+0.28%) | 68.38(-5.42%) | 50.53(-36.65%) | 36.81(-41.73%) | 47.29(-16.56%) | 52.06(-44.76%) | 56.33(+5.26%) |
| TIES | 69.12(-15.57%) | 36.38(+14.34%) | 68.38(-5.42%) | 53.99(-32.32%) | 36.81(-41.73%) | 49.81(-12.1%) | 50.91(-45.98%) | 45.07(-15.78%) |
| DARE | 69.12(-15.57%) | 32.73(+2.88%) | 68.38(-5.42%) | 49.46(-38.01%) | 36.81(-41.73%) | 52.7(-7.0%) | 50.91(-45.98%) | 43.66(-18.42%) |
| SLERP | 30.87(-62.29%) | 31.81(0.0%) | 68.38(-5.42%) | 50.53(-36.65%) | 36.81(-41.73%) | 47.29(-16.56%) | 49.08(-47.93%) | 56.33(+5.26%) |

## E.2 DETAILED EXPERIMENT RESULTS OF CONFLICT METRIC

Here we also provide the detailed data of merging pairs of Qwen2.5-3B checkpoints with different merging techniques in Table 18 and the calculated conflict metric in Table 19 ~ 22.

Table 19: Detail data of Parameter Magnitude Change Ratio between task pairs in GLUE dataset.

| | COLA | MNLI | MRPC | QNLI | QQP | RTE | SST2 | WNLI |
|---|---|---|---|---|---|---|---|---|
| COLA | - | 0.761 | 0.769 | 0.734 | 0.749 | 0.764 | 0.683 | 0.714 |
| MNLI | - | - | 0.691 | 0.717 | 0.714 | 0.687 | 0.752 | 0.744 |
| MRPC | - | - | - | 0.678 | 0.667 | 0.613 | 0.74 | 0.702 |
| QNLI | - | - | - | - | 0.717 | 0.68 | 0.735 | 0.695 |
| QQP | - | - | - | - | - | 0.696 | 0.751 | 0.718 |
| RTE | - | - | - | - | - | - | 0.745 | 0.667 |
| SST2 | - | - | - | - | - | - | - | 0.719 |
| WNLI | - | - | - | - | - | - | - | - |

Table 20: Detail data of Parameter Sign Change Ratio between task pairs in GLUE dataset.

| | COLA | MNLI | MRPC | QNLI | QQP | RTE | SST2 | WNLI |
|---|---|---|---|---|---|---|---|---|
| COLA | - | 0.106 | 0.109 | 0.096 | 0.121 | 0.106 | 0.081 | 0.101 |
| MNLI | - | - | 0.089 | 0.094 | 0.089 | 0.088 | 0.103 | 0.1 |
| MRPC | - | - | - | 0.089 | 0.1 | 0.075 | 0.101 | 0.092 |
| QNLI | - | - | - | - | 0.111 | 0.081 | 0.093 | 0.093 |
| QQP | - | - | - | - | - | 0.105 | 0.118 | 0.113 |
| RTE | - | - | - | - | - | - | 0.101 | 0.086 |
| SST2 | - | - | - | - | - | - | - | 0.1 |
| WNLI | - | - | - | - | - | - | - | - |

Table 18: Detail results of merging pairs of models' results among 8 fine-tuned Qwen2.5-3B checkpoints in GLUE dataset.

| LA | cola | mnli | mrpc | qnli | qqp | rte | sst2 | wnli |
|---|---|---|---|---|---|---|---|---|
| cola | - | -0.2 | -2.5 | -0.7 | -0.8 | -1.5 | -0.3 | -5.9 |
| mnli | -0.2 | - | -2.9 | -0.7 | 0.9 | -0.4 | 0.3 | -11.5 |
| mrpc | -2.5 | -2.9 | - | -2.5 | -2.2 | -5.9 | -2.2 | -9.7 |
| qnli | -0.7 | -0.7 | -2.5 | - | -4.8 | -2.8 | -0.5 | -12.6 |
| qqp | -0.8 | 0.9 | -2.2 | -4.8 | - | -2.3 | -0.6 | -9.8 |
| rte | -1.5 | -0.4 | -5.9 | -2.8 | -2.3 | - | -1.0 | -20.1 |
| sst2 | -0.3 | 0.3 | -2.2 | -0.5 | -0.6 | -1.0 | - | -9.0 |
| wnli | -5.9 | -11.5 | -9.7 | -12.6 | -9.8 | -20.1 | -9.0 | - |
| TA | cola | mnli | mrpc | qnli | qqp | rte | sst2 | wnli |
| cola | - | -0.1 | -2.1 | -0.5 | -1.2 | -1.6 | -0.4 | -8.0 |
| mnli | -0.1 | - | -2.8 | -0.7 | 0.9 | -0.4 | 0.3 | -11.5 |
| mrpc | -2.1 | -2.8 | - | -2.6 | -2.0 | -6.4 | -2.2 | -9.9 |
| qnli | -0.5 | -0.7 | -2.6 | - | -4.8 | -2.6 | -0.6 | -12.6 |
| qqp | -1.2 | 0.9 | -2.0 | -4.8 | - | -2.3 | -0.7 | -8.0 |
| rte | -1.6 | -0.4 | -6.4 | -2.6 | -2.3 | - | -0.9 | -20.1 |
| sst2 | -0.4 | 0.3 | -2.2 | -0.6 | -0.7 | -0.9 | - | -9.0 |
| wnli | -8.0 | -11.5 | -9.9 | -12.6 | -8.0 | -20.1 | -9.0 | - |
| TIES | cola | mnli | mrpc | qnli | qqp | rte | sst2 | wnli |
| cola | - | 0.3 | -1.9 | 0.1 | 0.2 | -2.0 | 0.2 | -1.1 |
| mnli | 0.3 | - | -2.3 | -0.4 | 1.0 | -0.4 | 0.7 | -12.3 |
| mrpc | -1.9 | -2.3 | - | -1.0 | -0.5 | -5.1 | -1.4 | -3.3 |
| qnli | 0.1 | -0.4 | -1.0 | - | -10.6 | -3.3 | -0.5 | -1.1 |
| qqp | 0.2 | 1.0 | -0.5 | -10.6 | - | -1.8 | -0.3 | -9.5 |
| rte | -2.0 | -0.4 | -5.1 | -3.3 | -1.8 | - | -0.3 | -24.7 |
| sst2 | 0.2 | 0.7 | -1.4 | -0.5 | -0.3 | -0.3 | - | -2.4 |
| wnli | -1.1 | -12.3 | -3.3 | -1.1 | -9.5 | -24.7 | -2.4 | - |
| DARE | cola | mnli | mrpc | qnli | qqp | rte | sst2 | wnli |
| cola | - | -1.1 | -2.5 | -0.1 | 1.4 | 0.4 | -1.4 | -0.4 |
| mnli | -1.1 | - | -2.5 | -1.2 | 1.8 | -0.2 | 0.9 | -16.6 |
| mrpc | -2.5 | -2.5 | - | 0.0 | -1.3 | -4.8 | -1.1 | -5.8 |
| qnli | -0.1 | -1.2 | 0.0 | - | -7.5 | -2.1 | -2.1 | -2.3 |
| qqp | 1.4 | 1.8 | -1.3 | -7.5 | - | -1.6 | 1.5 | -7.0 |
| rte | 0.4 | -0.2 | -4.8 | -2.1 | -1.6 | - | 0.0 | -25.7 |
| sst2 | -1.4 | 0.9 | -1.1 | -2.1 | 1.5 | 0.0 | - | -3.7 |
| wnli | -0.4 | -16.6 | -5.8 | -2.3 | -7.0 | -25.7 | -3.7 | - |
| SLERP | cola | mnli | mrpc | qnli | qqp | rte | sst2 | wnli |
| cola | - | -0.2 | -2.5 | -0.7 | -0.8 | -1.5 | -0.3 | -5.9 |
| mnli | -0.2 | - | -2.9 | -0.7 | 0.9 | -0.4 | 0.3 | -11.5 |
| mrpc | -2.5 | -2.9 | - | -2.5 | -2.2 | -5.9 | -47.3 | -9.7 |
| qnli | -0.7 | -0.7 | -2.5 | - | -4.8 | -2.8 | -0.5 | -12.6 |
| qqp | -0.8 | 0.9 | -2.2 | -4.8 | - | -2.3 | -0.6 | -9.8 |
| rte | -1.5 | -0.4 | -5.9 | -2.8 | -2.3 | - | -1.0 | -20.1 |
| sst2 | -0.3 | 0.3 | -47.3 | -0.5 | -0.6 | -1.0 | - | -9.0 |
| wnli | -5.9 | -11.5 | -9.7 | -12.6 | -9.8 | -20.1 | -9.0 | - |

Table 21: Detail data of Average cosine Similarity Between Pair's Parameters between task pairs in GLUE dataset.

| | COLA | MNLI | MRPC | QNLI | QQP | RTE | SST2 | WNLI |
|---|---|---|---|---|---|---|---|---|
| COLA | - | 0.277 | 0.284 | 0.253 | 0.292 | 0.279 | 0.212 | 0.258 |
| MNLI | - | - | 0.208 | 0.233 | 0.199 | 0.206 | 0.272 | 0.255 |
| MRPC | - | - | - | 0.212 | 0.218 | 0.167 | 0.267 | 0.211 |
| QNLI | - | - | - | - | 0.255 | 0.188 | 0.253 | 0.22 |
| QQP | - | - | - | - | - | 0.232 | 0.293 | 0.268 |
| RTE | - | - | - | - | - | - | 0.268 | 0.189 |
| SST2 | - | - | - | - | - | - | - | 0.26 |
| WNLI | - | - | - | - | - | - | - | - |

Table 22: Detail data of Average cosine Similarity Between Pair's Parameters between task pairs in GLUE dataset.

| | COLA | MNLI | MRPC | QNLI | QQP | RTE | SST2 | WNLI |
|---|---|---|---|---|---|---|---|---|
| COLA | - | 0.021 | 0.026 | 0.024 | 0.02 | 0.023 | 0.036 | 0.024 |
| MNLI | - | - | 0.036 | 0.032 | 0.03 | 0.038 | 0.019 | 0.025 |
| MRPC | - | - | - | 0.034 | 0.04 | 0.044 | 0.021 | 0.027 |
| QNLI | - | - | - | - | 0.026 | 0.035 | 0.02 | 0.028 |
| QQP | - | - | - | - | - | 0.032 | 0.017 | 0.023 |
| RTE | - | - | - | - | - | - | 0.02 | 0.036 |
| SST2 | - | - | - | - | - | - | - | 0.02 |
| WNLI | - | - | - | - | - | - | - | - |

We have also present our conflict metric data in a more intuitive way by drawing scatter graphs Figure 2, where we can directly find out that the weak correlation between the metric value and merging collapse.

### E.3  DETAILED EXPERIMENT RESULTS OF MERGING PERFORMANCE ON LOTS-OF-LORAS

In experiments on Lots-of-LoRAs, we randomly pick 64 rank=16 LoRA checkpoints and their corresponding evaluation datasets from Huggingface first, and pick 8 from them to form a merging group for 25 times. Which tasks our three example task groups (A), (B), (C) consist of is shown in Table 23:

We also draw heatmaps of similarity score in these 3 groups below in Figure 3 ~ 5  Additionally, we show our all 25 merging groups' task selection and detailed merging results in Table 24 ~ 48.

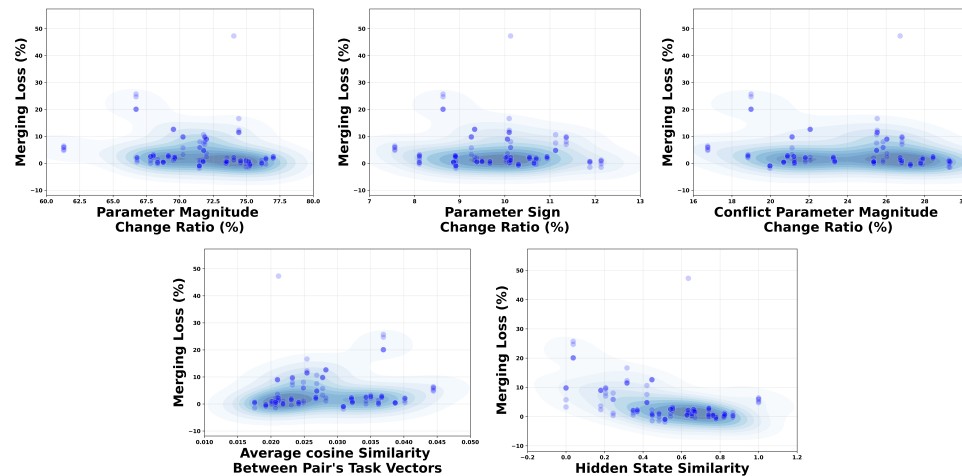

Figure 2: Scatter plot of merging loss and four types of parameter-level conflicts when merging pairs of Qwen2.5-3B checkpoints fine-tuned on 8 GLUE datasets.

Table 23: Lots-of-LoRAs tasks Used in Our Empirical Study.

| Merging Group | Tasks | |
|---|---|---|
| Group(a) | task1111_ted_translation_he_it, | task1243_ted_translation_gl_it |
| | task1331_reverse_array, | task290_tellmewhy_question_answerability |
| | task334_hateeval_classification_hate_es | task452_opus_paracrawl_en_ig_translation |
| | task531_europarl_es_en_translation | task592_sciq_incorrect_answer_generation |
| Group(b) | task1035_pib_translation_tamil_urdu | task1111_ted_translation_he_it |
| | task1145_xcsr_jap_commonsense_mc_classification | task1397_europa_ecdc_tm_fr_en_translation |
| | task175_spl_translation_en_pl | task452_opus_paracrawl_en_ig_translation |
| | task555_alt_translation_en_kh | task797_pawsx_spanish_french_translation |
| Group(c) | task1322_country_government_type | task1649_opus_books_en-no_translation |
| | task334_hateeval_classification_hate_es | task531_europarl_es_en_translation |
| | task741_lhoestq_answer_generation_place | task773_pawsx_spanish_text_modification |
| | task827_copa_commonsense_reasoning | task851_synthetic_multiply_evens |

Table 24: Detailed results of merging group (a).

| Method \ Task | 1111 | 1243 | 1331 | 290 | 334 | 452 | 531 | 592 |
|---|---|---|---|---|---|---|---|---|
| fine-tuned | 36.94 | 48.77 | 100.0 | 77.45 | 95.16 | 55.34 | 58.86 | 8.77 |
| LA | 33.37(-9.66%) | 45.46(-6.78%) | 99.35(-0.64%) | 90.63(+17.01%) | 90.5(-4.9%) | 12.16(-78.01%) | 56.18(-4.56%) | 7.32(-16.53%) |
| TA | 30.32(-17.91%) | 39.67(-18.65%) | 96.34(-3.65%) | 87.38(+12.81%) | 89.12(-6.34%) | 11.83(-78.61%) | 55.1(-6.39%) | 7.77(-11.4%) |
| TIES | 32.96(-10.75%) | 45.85(-5.98%) | 96.29(-3.7%) | 76.79(-0.86%) | 89.83(-5.6%) | 16.16(-70.79%) | 57.05(-3.07%) | 7.4(-15.52%) |

Table 25: Detailed results of merging group (b).

| Method \ Task | 1035 | 1111 | 1145 | 1397 | 175 | 452 | 555 | 797 |
|---|---|---|---|---|---|---|---|---|
| fine-tuned | 16.89 | 36.94 | 46.0 | 70.39 | 63.63 | 55.34 | 24.74 | 69.43 |
| LA | 17.3(+2.46%) | 33.56(-9.13%) | 32.18(-30.03%) | 62.13(-11.72%) | 56.81(-10.72%) | 13.41(-75.75%) | 14.78(-40.24%) | 69.95(+0.74%) |
| TA | 14.19(-15.97%) | 30.91(-16.3%) | 19.52(-57.54%) | 60.05(-14.68%) | 52.42(-17.61%) | 11.66(-78.92%) | 13.69(-44.65%) | 62.54(-9.91%) |
| TIES | 16.19(-4.13%) | 34.42(-6.8%) | 28.0(-39.13%) | 63.52(-9.75%) | 54.43(-14.45%) | 17.59(-68.2%) | 11.18(-54.78%) | 68.82(-0.88%) |

Table 26: Detailed results of merging group (c).

| Method \ Task | 1322 | 1649 | 334 | 531 | 741 | 773 | 827 | 851 |
|---|---|---|---|---|---|---|---|---|
| fine-tuned | 80.9 | 44.48 | 95.16 | 58.86 | 77.77 | 86.27 | 97.0 | 92.54 |
| LA | 65.57(-18.94%) | 38.86(-12.63%) | 91.41(-3.94%) | 55.95(-4.94%) | 72.85(-6.33%) | 77.18(-10.53%) | 96.0(-1.03%) | 13.39(-85.52%) |
| TA | 54.57(-32.54%) | 32.06(-27.9%) | 88.07(-7.45%) | 55.0(-6.55%) | 68.33(-12.13%) | 51.3(-40.52%) | 81.45(-16.02%) | 13.6(-85.3%) |
| TIES | 64.23(-20.6%) | 40.16(-9.7%) | 93.16(-2.1%) | 57.13(-2.93%) | 67.59(-13.08%) | 58.3(-32.42%) | 97.0(0.0%) | 18.21(-80.32%) |

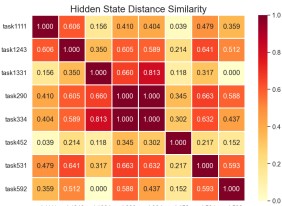
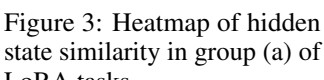

Figure 3: Heatmap of hidden state similarity in group (a) of LoRA tasks.

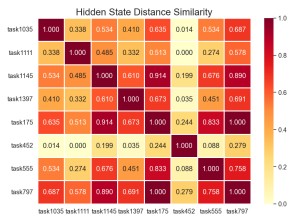

Figure 4: Heatmap of hidden state similarity in group (b) of LoRA tasks.

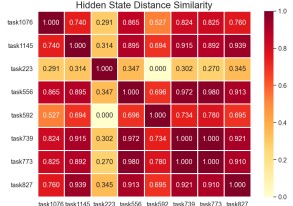

Figure 5: Heatmap of hidden state similarity in group (c) of LoRA tasks.

Table 27: Detailed results of merging group (d).

| Method \ Task | 105 | 110 | 1322 | 1423 | 1424 | 210 | 330 | 835 |
|---|---|---|---|---|---|---|---|---|
| fine-tuned | 23.67 | 99.89 | 80.9 | 22.81 | 25.86 | 99.94 | 91.08 | 50.45 |
| LA | 21.89(-7.52%) | 74.03(-25.88%) | 63.88(-21.03%) | 26.61(+16.66%) | 29.31(+13.33%) | 88.92(-11.03%) | 83.99(-7.78%) | 31.8(-36.97%) |
| TA | 21.12(-10.77%) | 65.14(-34.78%) | 62.18(-23.13%) | 15.07(-33.91%) | 13.32(-48.46%) | 77.23(-22.72%) | 80.71(-11.37%) | 23.57(-53.27%) |
| TIES | 22.89(-3.29%) | 81.37(-18.53%) | 61.11(-24.46%) | 22.43(-1.66%) | 27.58(+6.66%) | 93.09(-6.85%) | 87.19(-4.26%) | 30.11(-40.31%) |

Table 28: Detailed results of merging group (e).

| Method \ Task | 132 | 1577 | 253 | 440 | 797 | 827 | 861 | 909 |
|---|---|---|---|---|---|---|---|---|
| fine-tuned | 100.0 | 100.0 | 90.27 | 98.61 | 69.43 | 97.0 | 63.05 | 79.54 |
| LA | 98.71(-1.28%) | 100.0(0.0%) | 86.57(-4.1%) | 69.02(-30.0%) | 67.91(-2.19%) | 94.0(-3.09%) | 69.33(+9.95%) | 68.18(-14.28%) |
| TA | 92.63(-7.36%) | 100.0(0.0%) | 74.77(-17.17%) | 46.3(-53.04%) | 61.22(-11.82%) | 85.53(-11.82%) | 68.54(+8.71%) | 57.74(-27.4%) |
| TIES | 99.89(-0.1%) | 100.0(0.0%) | 83.79(-7.17%) | 94.3(-4.36%) | 68.65(-1.11%) | 95.0(-2.06%) | 63.16(+0.16%) | 72.72(-8.57%) |

Table 29: Detailed results of merging group (f).

| Method \ Task | 1375 | 175 | 335 | 399 | 739 | 827 | 839 | 984 |
|---|---|---|---|---|---|---|---|---|
| fine-tuned | 51.92 | 63.63 | 90.77 | 94.25 | 18.01 | 97.0 | 90.61 | 5.55 |
| LA | 48.37(-6.84%) | 61.13(-3.92%) | 69.27(-23.68%) | 93.56(-0.73%) | 19.43(+7.88%) | 94.0(-3.09%) | 86.47(-4.57%) | 3.82(-31.19%) |
| TA | 44.93(-13.45%) | 51.06(-19.75%) | 64.93(-28.47%) | 88.77(-5.81%) | 24.46(+35.82%) | 82.64(-14.79%) | 68.33(-24.59%) | 0.64(-88.31%) |
| TIES | 48.08(-7.39%) | 54.0(-15.13%) | 90.88(+0.12%) | 92.98(-1.34%) | 17.18(-4.59%) | 94.0(-3.09%) | 88.61(-2.2%) | 4.75(-14.34%) |

Table 30: Detailed results of merging group (g).

| Method \ Task | 1035 | 1076 | 1240 | 1497 | 281 | 305 | 544 | 556 |
|---|---|---|---|---|---|---|---|---|
| fine-tuned | 16.89 | 14.85 | 62.59 | 95.83 | 68.47 | 64.76 | 50.77 | 32.38 |
| LA | 17.57(+4.06%) | 15.59(+4.99%) | 59.91(-4.28%) | 85.67(-10.59%) | 51.85(-24.27%) | 61.68(-4.76%) | 44.74(-11.86%) | 29.02(-10.37%) |
| TA | 12.9(-23.6%) | 13.65(-8.1%) | 52.38(-16.3%) | 64.09(-33.11%) | 50.74(-25.89%) | 53.85(-16.85%) | 43.12(-15.05%) | 20.85(-35.62%) |
| TIES | 15.29(-9.45%) | 13.31(-10.38%) | 61.36(-1.96%) | 85.41(-10.86%) | 29.38(-57.09%) | 64.9(+0.2%) | 42.17(-16.93%) | 23.95(-26.04%) |

Table 31: Detailed results of merging group (h).

| Method \ Task | 1329 | 547 | 555 | 773 | 808 | 851 | 872 | 873 |
|---|---|---|---|---|---|---|---|---|
| fine-tuned | 1.02 | 99.87 | 24.74 | 86.27 | 57.83 | 92.54 | 9.16 | 26.52 |
| LA | 0.87(-14.68%) | 98.99(-0.87%) | 15.56(-37.08%) | 83.82(-2.83%) | 47.17(-18.43%) | 12.59(-86.39%) | 1.16(-87.34%) | 15.16(-42.85%) |
| TA | 0.37(-63.45%) | 93.09(-6.78%) | 10.18(-58.83%) | 61.08(-29.19%) | 46.9(-18.88%) | 13.21(-85.71%) | 1.08(-88.13%) | 13.71(-48.28%) |
| TIES | 1.04(+1.11%) | 98.66(-1.2%) | 12.76(-48.42%) | 83.3(-3.44%) | 35.33(-38.9%) | 13.72(-85.17%) | 1.76(-80.71%) | 18.69(-29.52%) |

Table 32: Detailed results of merging group (i).

| Method \ Task | 1145 | 1397 | 397 | 416 | 537 | 573 | 688 | 741 |
|---|---|---|---|---|---|---|---|---|
| fine-tuned | 46.0 | 70.39 | 93.74 | 30.31 | 48.28 | 99.81 | 50.0 | 77.77 |
| LA | 34.0(-26.08%) | 62.41(-11.33%) | 91.69(-2.17%) | 27.01(-10.89%) | 44.35(-8.12%) | 99.81(0.0%) | 41.66(-16.66%) | 77.16(-0.78%) |
| TA | 29.64(-35.54%) | 59.29(-15.75%) | 87.31(-6.86%) | 24.67(-18.58%) | 43.18(-10.55%) | 98.7(-1.1%) | 41.77(-16.45%) | 71.97(-7.46%) |
| TIES | 32.0(-30.43%) | 65.46(-6.99%) | 92.84(-0.95%) | 29.71(-1.97%) | 43.41(-10.08%) | 99.81(0.0%) | 50.0(0.0%) | 74.59(-4.08%) |

Table 33: Detailed results of merging group (j).

| Method \ Task | 105 | 1145 | 1424 | 1497 | 1577 | 556 | 592 | 835 |
|---|---|---|---|---|---|---|---|---|
| fine-tuned | 23.67 | 46.0 | 25.86 | 95.83 | 100.0 | 32.38 | 8.77 | 50.45 |
| LA | 21.35(-9.81%) | 35.0(-23.91%) | 29.31(+13.33%) | 87.5(-8.69%) | 100.0(0.0%) | 31.25(-3.48%) | 8.02(-8.45%) | 35.12(-30.38%) |
| TA | 21.21(-10.4%) | 32.89(-28.48%) | 12.08(-53.28%) | 78.67(-17.9%) | 100.0(0.0%) | 24.66(-23.84%) | 7.95(-9.25%) | 23.99(-52.44%) |
| TIES | 22.87(-3.39%) | 30.0(-34.78%) | 32.75(+26.66%) | 87.5(-8.69%) | 81.81(-18.18%) | 29.79(-8.01%) | 7.63(-12.9%) | 39.13(-22.43%) |

Table 34: Detailed results of merging group (k).

| Task
Method | 1035 | 110 | 210 | 253 | 281 | 555 | 688 | 797 |
|---|---|---|---|---|---|---|---|---|
| fine-tuned | 16.89 | 99.89 | 99.94 | 90.27 | 68.47 | 24.74 | 50.0 | 69.43 |
| LA | 17.37(+2.86%) | 77.33(-22.58%) | 89.22(-10.72%) | 83.84(-7.12%) | 55.2(-19.38%) | 9.77(-60.48%) | 41.66(-16.66%) | 69.35(-0.11%) |
| TA | 13.5(-20.07%) | 64.38(-35.54%) | 74.29(-25.67%) | 75.88(-15.94%) | 48.96(-28.49%) | 12.78(-48.34%) | 24.4(-51.18%) | 57.99(-16.47%) |
| TIES | 16.66(-1.36%) | 90.33(-9.56%) | 99.23(-0.71%) | 87.5(-3.07%) | 55.91(-18.34%) | 15.55(-37.13%) | 33.33(-33.33%) | 69.72(+0.41%) |

Table 35: Detailed results of merging group (l).

| Task
Method | 132 | 1397 | 397 | 440 | 544 | 547 | 739 | 839 |
|---|---|---|---|---|---|---|---|---|
| fine-tuned | 100.0 | 70.39 | 93.74 | 98.61 | 50.77 | 99.87 | 18.01 | 90.61 |
| LA | 99.24(-0.75%) | 61.9(-12.04%) | 91.06(-2.86%) | 54.0(-45.23%) | 46.89(-7.62%) | 97.56(-2.31%) | 21.78(+20.92%) | 86.92(-4.07%) |
| TA | 92.82(-7.17%) | 59.59(-15.34%) | 85.62(-8.66%) | 46.39(-52.95%) | 44.73(-11.88%) | 93.49(-6.38%) | 23.3(+29.39%) | 73.0(-19.43%) |
| TIES | 99.77(-0.22%) | 65.37(-7.12%) | 91.57(-2.31%) | 91.84(-6.86%) | 42.52(-16.23%) | 97.67(-2.2%) | 17.11(-4.97%) | 88.3(-2.54%) |

Table 36: Detailed results of merging group (m).

| Task
Method | 1129 | 1240 | 1329 | 1375 | 330 | 409 | 873 | 909 |
|---|---|---|---|---|---|---|---|---|
| fine-tuned | 99.84 | 62.59 | 1.02 | 51.92 | 91.08 | 31.77 | 26.52 | 79.54 |
| LA | 97.4(-2.44%) | 60.81(-2.83%) | 0.78(-23.88%) | 48.06(-7.43%) | 84.77(-6.92%) | 30.43(-4.23%) | 14.97(-43.56%) | 69.69(-12.38%) |
| TA | 57.61(-42.29%) | 56.11(-10.35%) | 0.5(-51.2%) | 46.49(-10.46%) | 81.75(-10.23%) | 28.24(-11.09%) | 13.89(-47.6%) | 51.51(-35.23%) |
| TIES | 94.75(-5.1%) | 60.94(-2.62%) | 0.86(-16.11%) | 47.55(-8.41%) | 87.89(-3.5%) | 31.56(-0.66%) | 16.64(-37.24%) | 58.33(-26.66%) |

Table 37: Detailed results of merging group (n).

| Task
Method | 1076 | 1107 | 1423 | 223 | 399 | 416 | 573 | 827 |
|---|---|---|---|---|---|---|---|---|
| fine-tuned | 14.85 | 28.86 | 22.81 | 73.34 | 94.25 | 30.31 | 99.81 | 97.0 |
| LA | 14.82(-0.23%) | 25.37(-12.06%) | 28.7(+25.83%) | 27.34(-62.71%) | 93.33(-0.97%) | 27.72(-8.53%) | 99.81(0.0%) | 97.0(0.0%) |
| TA | 12.09(-18.57%) | 22.66(-21.47%) | 11.62(-49.05%) | 23.91(-67.38%) | 89.71(-4.81%) | 25.2(-16.85%) | 99.08(-0.72%) | 88.94(-8.3%) |
| TIES | 0.8(-94.61%) | 15.16(-47.44%) | 21.67(-5.0%) | 30.52(-58.37%) | 91.72(-2.68%) | 29.67(-2.1%) | 99.81(0.0%) | 88.0(-9.27%) |

Table 38: Detailed results of merging group (o).

| Task
Method | 1111 | 175 | 290 | 644 | 808 | 861 | 872 | 984 |
|---|---|---|---|---|---|---|---|---|
| fine-tuned | 36.94 | 63.63 | 77.45 | 65.8 | 57.83 | 63.05 | 9.16 | 5.55 |
| LA | 33.58(-9.09%) | 56.8(-10.74%) | 90.63(+17.01%) | 65.23(-0.87%) | 54.73(-5.35%) | 63.23(+0.27%) | 0.83(-90.84%) | 5.55(0.0%) |
| TA | 30.82(-16.56%) | 54.01(-15.11%) | 83.79(+8.17%) | 64.41(-2.11%) | 48.99(-15.28%) | 61.61(-2.27%) | 1.08(-88.18%) | 1.71(-69.13%) |
| TIES | 33.92(-8.17%) | 54.98(-13.6%) | 92.83(+19.85%) | 64.66(-1.74%) | 43.76(-24.31%) | 64.19(+1.81%) | 1.07(-88.22%) | 6.11(+10.0%) |

Table 39: Detailed results of merging group (p).

| Task
Method | 114 | 1243 | 1423 | 253 | 644 | 827 | 839 | 984 |
|---|---|---|---|---|---|---|---|---|
| fine-tuned | 99.38 | 48.77 | 22.81 | 90.27 | 65.8 | 97.0 | 90.61 | 5.55 |
| LA | 82.46(-17.02%) | 45.01(-7.71%) | 27.37(+20.0%) | 84.16(-6.76%) | 65.89(+0.13%) | 95.0(-2.06%) | 87.07(-3.9%) | 4.75(-14.37%) |
| TA | 26.19(-73.63%) | 41.24(-15.44%) | 10.65(-53.28%) | 74.12(-17.88%) | 63.17(-4.0%) | 90.4(-6.8%) | 75.88(-16.25%) | 1.76(-68.25%) |
| TIES | 92.92(-6.5%) | 44.49(-8.78%) | 22.81(0.0%) | 79.14(-12.33%) | 63.99(-2.76%) | 77.0(-20.61%) | 86.92(-4.07%) | 2.71(-51.19%) |

Table 40: Detailed results of merging group (q).

| Task
Method | 1076 | 1145 | 223 | 556 | 592 | 739 | 773 | 827 |
|---|---|---|---|---|---|---|---|---|
| fine-tuned | 14.85 | 46.0 | 73.34 | 32.38 | 8.77 | 18.01 | 86.27 | 97.0 |
| LA | 15.28(+2.9%) | 39.0(-15.21%) | 27.24(-62.84%) | 26.37(-18.57%) | 7.62(-13.1%) | 16.96(-5.82%) | 79.07(-8.33%) | 96.0(-1.03%) |
| TA | 13.77(-7.25%) | 27.25(-40.74%) | 23.73(-67.64%) | 25.15(-22.34%) | 7.85(-10.44%) | 22.74(+26.26%) | 55.65(-35.48%) | 81.77(-15.69%) |
| TIES | 14.84(-0.08%) | 31.0(-32.6%) | 31.49(-57.05%) | 30.48(-5.88%) | 7.47(-14.73%) | 19.19(+6.56%) | 82.4(-4.48%) | 96.0(-1.03%) |

Table 41: Detailed results of merging group (r).

| Task
Method | 1035 | 132 | 1577 | 1649 | 573 | 688 | 741 | 808 |
|---|---|---|---|---|---|---|---|---|
| fine-tuned | 16.89 | 100.0 | 100.0 | 44.48 | 99.81 | 50.0 | 77.77 | 57.83 |
| LA | 17.41(+3.11%) | 99.11(-0.88%) | 100.0(0.0%) | 39.32(-11.58%) | 99.81(0.0%) | 41.66(-16.66%) | 74.39(-4.35%) | 48.25(-16.55%) |
| TA | 14.23(-15.7%) | 93.35(-6.64%) | 100.0(0.0%) | 33.04(-25.7%) | 97.23(-2.57%) | 33.33(-33.33%) | 72.43(-6.87%) | 47.69(-17.52%) |
| TIES | 18.64(+10.37%) | 99.93(-0.06%) | 100.0(0.0%) | 41.22(-7.31%) | 99.81(0.0%) | 50.0(0.0%) | 78.51(+0.95%) | 50.13(-13.31%) |

Table 42: Detailed results of merging group (s).

| Task
Method | 105 | 1240 | 1331 | 1424 | 1497 | 409 | 797 | 872 |
|---|---|---|---|---|---|---|---|---|
| fine-tuned | 23.67 | 62.59 | 100.0 | 25.86 | 95.83 | 31.77 | 69.43 | 9.16 |
| LA | 21.86(-7.67%) | 61.0(-2.53%) | 99.1(-0.89%) | 27.58(+6.66%) | 86.8(-9.42%) | 30.58(-3.74%) | 69.9(+0.67%) | 0.91(-90.04%) |
| TA | 21.36(-9.77%) | 55.84(-10.78%) | 96.4(-3.59%) | 4.44(-82.8%) | 72.27(-24.58%) | 28.03(-11.77%) | 64.06(-7.73%) | 1.04(-88.63%) |
| TIES | 23.33(-1.46%) | 61.96(-1.0%) | 97.58(-2.41%) | 29.31(+13.33%) | 84.72(-11.59%) | 31.56(-0.66%) | 69.24(-0.27%) | 1.7(-81.43%) |

Table 43: Detailed results of merging group (t).

| Task
Method | 175 | 210 | 290 | 397 | 440 | 537 | 861 | 873 |
|---|---|---|---|---|---|---|---|---|
| fine-tuned | 63.63 | 99.94 | 77.45 | 93.74 | 98.61 | 48.28 | 63.05 | 26.52 |
| LA | 55.53(-12.72%) | 84.42(-15.52%) | 90.59(+16.95%) | 92.84(-0.95%) | 51.2(-48.07%) | 45.33(-6.09%) | 63.32(+0.43%) | 14.65(-44.76%) |
| TA | 51.39(-19.23%) | 74.17(-25.78%) | 87.04(+12.38%) | 87.57(-6.58%) | 46.02(-53.33%) | 43.54(-9.81%) | 68.74(+9.02%) | 13.23(-50.11%) |
| TIES | 49.09(-22.85%) | 95.47(-4.47%) | 93.12(+20.22%) | 93.48(-0.27%) | 82.64(-16.19%) | 40.96(-15.14%) | 53.34(-15.39%) | 16.94(-36.14%) |

Table 44: Detailed results of merging group (u).

| Task
Method | 110 | 1375 | 281 | 305 | 399 | 835 | 851 | 909 |
|---|---|---|---|---|---|---|---|---|
| fine-tuned | 99.89 | 51.92 | 68.47 | 64.76 | 94.25 | 50.45 | 92.54 | 79.54 |
| LA | 73.01(-26.9%) | 48.71(-6.18%) | 54.67(-20.15%) | 62.0(-4.27%) | 93.67(-0.6%) | 28.21(-44.08%) | 17.23(-81.37%) | 68.18(-14.28%) |
| TA | 62.66(-37.26%) | 44.92(-13.48%) | 50.84(-25.74%) | 56.3(-13.06%) | 85.86(-8.9%) | 14.88(-70.49%) | 14.0(-84.86%) | 56.81(-28.57%) |
| TIES | 84.95(-14.95%) | 47.14(-9.2%) | 41.57(-39.27%) | 63.04(-2.66%) | 93.1(-1.21%) | 32.49(-35.59%) | 15.45(-83.3%) | 62.12(-21.9%) |

Table 45: Detailed results of merging group (v).

| Task
Method | 1107 | 1129 | 1322 | 330 | 334 | 335 | 547 | 555 |
|---|---|---|---|---|---|---|---|---|
| fine-tuned | 28.86 | 99.84 | 80.9 | 91.08 | 95.16 | 90.77 | 99.87 | 24.74 |
| LA | 22.84(-20.84%) | 98.0(-1.84%) | 65.34(-19.22%) | 84.77(-6.92%) | 90.41(-4.99%) | 77.55(-14.56%) | 92.8(-7.07%) | 17.4(-29.65%) |
| TA | 19.75(-31.55%) | 57.34(-42.56%) | 56.23(-30.49%) | 80.35(-11.77%) | 88.9(-6.58%) | 68.03(-25.04%) | 90.48(-9.4%) | 12.88(-47.93%) |
| TIES | 18.8(-34.83%) | 96.29(-3.55%) | 66.73(-17.51%) | 89.1(-2.17%) | 90.75(-4.64%) | 91.22(+0.48%) | 92.96(-6.91%) | 10.65(-56.92%) |

Table 46: Detailed results of merging group (w).

| Task
Method | 1076 | 1329 | 1331 | 397 | 531 | 547 | 835 | 861 |
|---|---|---|---|---|---|---|---|---|
| fine-tuned | 14.85 | 1.02 | 100.0 | 93.74 | 58.86 | 99.87 | 50.45 | 63.05 |
| LA | 16.09(+8.35%) | 0.68(-33.73%) | 99.23(-0.76%) | 91.12(-2.79%) | 56.0(-4.85%) | 94.32(-5.54%) | 32.2(-36.16%) | 70.7(+12.12%) |
| TA | 13.27(-10.61%) | 0.64(-37.29%) | 96.42(-3.57%) | 83.39(-11.03%) | 55.01(-6.54%) | 90.93(-8.94%) | 17.04(-66.22%) | 67.41(+6.91%) |
| TIES | 13.65(-8.11%) | 1.2(+16.66%) | 99.35(-0.64%) | 93.74(0.0%) | 57.11(-2.97%) | 95.5(-4.37%) | 39.15(-22.4%) | 68.71(+8.98%) |

Table 47: Detailed results of merging group (x).

| Task
Method | 1107 | 1240 | 1375 | 1577 | 537 | 573 | 808 | 909 |
|---|---|---|---|---|---|---|---|---|
| fine-tuned | 28.86 | 62.59 | 51.92 | 100.0 | 48.28 | 99.81 | 57.83 | 79.54 |
| LA | 26.99(-6.47%) | 60.3(-3.65%) | 49.32(-4.99%) | 100.0(0.0%) | 43.72(-9.43%) | 99.81(0.0%) | 42.57(-26.38%) | 65.9(-17.14%) |
| TA | 24.02(-16.74%) | 54.1(-13.56%) | 46.03(-11.33%) | 100.0(0.0%) | 43.24(-10.43%) | 96.89(-2.91%) | 48.38(-16.33%) | 57.81(-27.31%) |
| TIES | 27.77(-3.76%) | 61.44(-1.84%) | 49.77(-4.14%) | 78.78(-21.21%) | 42.16(-12.67%) | 99.81(0.0%) | 37.79(-34.64%) | 72.72(-8.57%) |

Table 48: Detailed results of merging group (y).

| Task
Method | 1424 | 1497 | 253 | 305 | 330 | 335 | 409 | 544 |
|---|---|---|---|---|---|---|---|---|
| fine-tuned | 25.86 | 95.83 | 90.27 | 64.76 | 91.08 | 90.77 | 31.77 | 50.77 |
| LA | 27.58(+6.66%) | 88.19(-7.97%) | 81.01(-10.25%) | 62.42(-3.61%) | 84.16(-7.59%) | 76.83(-15.36%) | 29.44(-7.32%) | 45.73(-9.91%) |
| TA | 9.96(-61.48%) | 74.28(-22.48%) | 70.23(-22.19%) | 57.69(-10.92%) | 81.38(-10.65%) | 68.79(-24.21%) | 27.74(-12.68%) | 44.36(-12.6%) |
| TIES | 32.75(+26.66%) | 87.5(-8.69%) | 81.01(-10.25%) | 62.58(-3.37%) | 88.76(-2.54%) | 89.55(-1.34%) | 30.89(-2.77%) | 40.3(-20.6%) |

We also present a line chart Figure 6 of the results on Lots-of-LoRAs dataset for intuitive understanding.

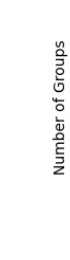
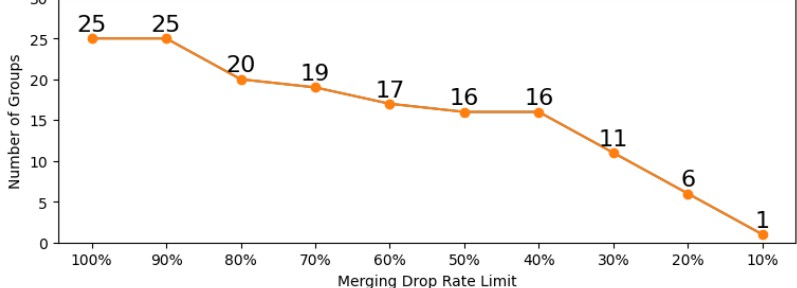

Figure 6: Numbers of groups with maximal drop rate not exceeding Merging Drop Rate Limit.

### E.4 MERGING TASKS SELECTION GUIDING.

Merging models with higher similarity scores could lead to better merging performance. In group (a), the 6-th task incurs significant merging collapse. Using our MDS metric as a guide, we strategically replace this task with more compatible alternatives from Lots-of-LoRA tasks and construct task gropus (a1) and (a2). As expected, we find merging tasks with lower MDS do achieve lesser merging collapse, which shows that hidden state similarity correlates merging collapse stronger and further supports our theory explanation and. Here, we show detail data in Table 49 and 50 with merging loss and MDS in the two new added group.

Table 49: Detail data when merging group (a1) tasks from Lots-of-LoRAs.

|       | task1111 | task1243 | task1331 | task290 | task334 | task335 | task531 | task592 |
|-------|----------|----------|----------|---------|---------|---------|---------|---------|
| LA    | -11.8    | -9.6     | -0.0     | 18.2    | -4.8    | -23.7   | -4.5    | -18.0   |
| TIES  | -13.9    | -7.6     | -3.0     | 14.8    | -6.0    | -17.7   | -4.1    | -16.2   |
| MDS   | 2.519    | 1.822    | 2.321    | 1.449   | 1.446   | 1.646   | 1.798   | 2.449   |

Table 50: Detail data when merging group (a2) tasks from Lots-of-LoRAs.

|       | task1111 | task1243 | task1331 | task290 | task334 | task440 | task531 | task592 |
|-------|----------|----------|----------|---------|---------|---------|---------|---------|
| LA    | -14.0    | -8.2     | -0.1     | 18.2    | -4.3    | -49.8   | -4.8    | -15.5   |
| TIES  | -12.5    | -7.6     | -0.4     | 16.2    | -3.5    | -19.1   | -4.2    | -18.5   |
| MDS   | 2.891    | 1.986    | 2.592    | 1.512   | 1.533   | 2.130   | 1.845   | 2.507   |

## F PARAMETER TUNING OF MERGING TECHNIQUES

### F.1 PARAMETER TUNING METHOD

- **8-to-1 Grid Search:** GLUE tasks with density/lambda tuning
    - TIES: density 0.1–0.8
    - DARE: density 0.1–0.8
    - TA: lambda (overall scaling factor) 0.1–0.8
- **Pairwise Grid Search:** optimization on task pairs exhibiting task-level collapse
    - Both density/lambda and weight tuning
    - Increment 0.1 for parameters
- **AdaMerging Yang et al. (2023):** learning-based optimization on task pairs exhibiting task-level collapse
    - 500 epochs, learning rate = $1 \times 10^{-3}$
    - 16 samples per epoch

We show the **merging loss (%)** of our default setting (used in our paper) and best results (the lowest average merging loss across tasks under hyperparameter tuning) in the following tables.

### F.2 EXPERIMENT RESULTS

Even with hyperparameter optimization, task-level incompatibility patterns remain consistent despite the overall merging loss being smaller on average. Notably, TIES and DARE still perform worse than the simple Linear Average on QNLI in 8-to-1 Grid Search experiment even with hyperparameter tuning. These results strengthen our claim that collapse stems from task-level incompatibility rather than suboptimal merging configurations.

Table 51: Experiment 1: 8-to-1 Grid Search on GLUE

|  | COLA | MNLI | MRPC | QNLI | QQP | RTE | SST2 | WNLI |
|---|---|---|---|---|---|---|---|---|
| TIES (best) | -4.4 | -0.2 | -7.4 | -24.3 | -21.9 | -2.4 | -2.5 | -36.5 |
| TIES (default) | -4.3 | -0.7 | -6.6 | -26.9 | -24.9 | -2.8 | -3.7 | -36.5 |
| DARE (best) | -4.5 | -0.3 | -7.9 | -23.5 | -24.5 | -2.4 | -2.5 | -34.5 |
| DARE (default) | -14.9 | -60.0 | -7.7 | -45.4 | -24.9 | -10.4 | -22.7 | -30.8 |
| TA (best) | -1.8 | -2.5 | -15.1 | -7.9 | -25.7 | -4.4 | -1.9 | -48.0 |
| TA (default) | -1.9 | -2.5 | -15.6 | -8.1 | -25.7 | -5.2 | -1.9 | -48.1 |

Table 52: Experiment 2: Pairwise Grid
Search with Weight Optimization

| **RTE+WNLI** | best | default |
|---|---|---|
| TIES | -40.2 | -49.46 |
| DARE | -36.2 | -50.46 |
| TA | -40.52 | -40.18 |
| **MNLI+WNLI** | best | default |
| TIES | -22.02 | -24.54 |
| DARE | -24.54 | -33.26 |
| TA | -21.84 | -22.94 |

Table 53: Experiment 3: AdaMerging Optimization

| **RTE+WNLI** | best | default |
|---|---|---|
| ADA ($\lambda_1 = 0.1261, \lambda_2 = 0.3216$) | -32.01 | -40.18 |
| **MNLI+WNLI** | best | default |
| ADA ($\lambda_1 = 0.4991, \lambda_2 = 0.0$) | -62.86 | -22.94 |

## G GENERALIZATION EXPERIMENTS

To further validate the prevalence of merging collapse, we perform following experiments additionally.

### G.1 CODING TASKS

We conduct additional experiments on CodeXGLUE Lu et al. (2021), a more challenging and widely used benchmark in the code domain. We fine-tune Qwen2.5-7B models on BigCloneBench, Devign, MicrosoftDocs, and CodeSearchNet, which are tasks within CodeXGLUE. The experiment results below imply that our analyses generalize:

Table 54: Performance comparison on CodeXGLUE tasks

| Merging | BigCloneBench | | Devign | | MicrosoftDocs | | CodeSearchNet | |
|---|---|---|---|---|---|---|---|---|
|  | FT | M($\Delta$) | FT | M($\Delta$) | FT | M($\Delta$) | FT | M($\Delta$) |
| TA | 41.2 | 19.2 (-53.5) | 26.3 | 4.0 (-85.0) | 67.8 | 61.9 (-8.6) | 98.1 | 90.8 (-7.4) |
| TIES | 41.2 | 19.0 (-53.8) | 26.3 | 3.1 (-88.3) | 67.8 | 59.7 (-11.9) | 98.1 | 93.9 (-4.2) |
| DARE | 41.2 | 10.8 (-73.7) | 26.3 | 4.7 (-82.1) | 67.8 | 65.2 (-3.8) | 98.1 | 97.9 (-0.1) |

FT: Fine-tuned model performance; $M(\Delta)$: Performance
after merging and the negative number of merging loss fol-
lowing Equation 1.

Table 55: ANOVA F-test results (p-values)

|  | Merging-Technique-Level | Task-Level |
|---|---|---|
| GLUE Tasks | 0.57 | $2.4 \times 10^{-36} \ll 0.05$ |
| Code Tasks | 0.97 | $2.8 \times 10^{-5} \ll 0.05$ |

The ANOVA F-tests on GLUE tasks and Code tasks demonstrate the same task-level collapse pattern, confirming that our analysis extends beyond simple NLP tasks to more challenging code tasks and our findings generalize well to more complex domains. We will add these analysis results in revision.

## G.2 ROBERTA MODEL

We conduct experiments on RoBERTa Liu et al. (2019) models. We made our best efforts to merge publicly available RoBERTa checkpoints from Huggingface using existing merging methods. The results consistently show similar collapse patterns reflected by the statistical tests shown in the table below, further validating our method's generalization across different language model families.

Table 56: Performance comparison on RoBERTa models across GLUE tasks

| Method | COLA | | MNLI | | MRPC | | QNLI | | QQP | | RTE | | SST-2 | | WNLI | |
|---|---|---|---|---|---|---|---|---|---|---|---|---|---|---|---|---|
|  | FT | M($\Delta$) | FT | M($\Delta$) | FT | M($\Delta$) | FT | M($\Delta$) | FT | M($\Delta$) | FT | M($\Delta$) | FT | M($\Delta$) | FT | M($\Delta$) |
| LA | 84.6 | 69.1 (-18.3) | 87.1 | 31.9 (-63.3) | 90.2 | 67.9 (-24.7) | 92.4 | 50.6 (-45.3) | 73.1 | 39.9 (-45.5) | 79.4 | 47.3 (-40.5) | 93.6 | 50.9 (-45.6) | 56.3 | 43.7 (-22.5) |
| TA | 84.6 | 30.9 (-63.5) | 87.1 | 33.9 (-61.1) | 90.2 | 31.6 (-64.9) | 92.4 | 49.5 (-46.5) | 73.1 | 62.9 (-14.0) | 79.4 | 52.7 (-33.6) | 93.6 | 49.1 (-47.5) | 56.3 | 56.3 (-0.0) |
| TIES | 84.6 | 30.9 (-63.5) | 87.1 | 35.3 (-59.4) | 90.2 | 31.6 (-64.9) | 92.4 | 49.5 (-46.5) | 73.1 | 63.2 (-13.6) | 79.4 | 52.7 (-33.6) | 93.6 | 49.1 (-47.5) | 56.3 | 56.3 (-0.0) |

Table 57: ANOVA F-test results (p-values) across different model families

| Model | Merging-Technique-Level | Task-Level |
|---|---|---|
| Qwen-2.5 3B | 0.57 | $2.4 \times 10^{-36} \ll 0.05$ |
| RoBERTa | 0.932991 | $6.69 \times 10^{-4} \ll 0.05$ |

## G.3 EMR-MERGING

We conducted experiments with EMRMerging Huang et al. (2024) and Qwen-2.5-3B on GLUE tasks, and the results shown in the table below complies with our core findings.

Table 58: EMRMerging performance on GLUE tasks with Qwen-2.5-3B

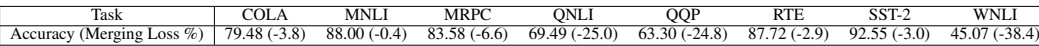

| Task | COLA | MNLI | MRPC | QNLI | QQP | RTE | SST-2 | WNLI |
|---|---|---|---|---|---|---|---|---|
| Accuracy (Merging Loss %) | 79.48 (-3.8) | 88.00 (-0.4) | 83.58 (-6.6) | 69.49 (-25.0) | 63.30 (-24.8) | 87.72 (-2.9) | 92.55 (-3.0) | 45.07 (-38.4) |

Specifically, we observe that the same task-level collapse pattern occurs with EMRMerging, despite the overall merging loss being smaller on average. This cross-method consistency reinforces our claim that task-level collapse is a fundamental phenomenon in model merging rather than a method-specific artifact.

