# OpenReview forum: "An Empirical Study and Theoretical Explanation on Task-Level Model-Merging Collapse"
_ICLR.cc/2026/Conference — Submitted to ICLR 2026_

### Official Review · Reviewer_sAmL · 2025-10-30

**Soundness:** 3
**Presentation:** 3
**Contribution:** 3
**Rating:** 6
**Confidence:** 3

**Summary:**

Model merging is a popular method to efficiently build a multi-task model. However, it usually leads to a performance degradation. In this paper, they propose three key research questions, including the consistency of performance degradation across merging techniques, the potential reason for performance degradation, and the factors affecting the merging performance. They conduct extensive experiments to answer those questions. Besides, a theoretical framework is proposed to analyze the relationship between representational incompatibility and merging collapse.

**Strengths:**

- The experiments in this paper are comprehensive, including diverse models and datasets.
- Besides empirical results, this paper proposed a theoretical framework to analyze the factors for merging collapse, revealing more insights to understand it.
- The research questions proposed in this paper are valuable and interesting, and the author gives a clear answer supported by extensive experiments.

**Weaknesses:**

- This paper considers the decoder-only models and encoder-decoder models, so it would be better to consider encoder-only models. An interesting direction would be investigating the effect of different model architectures.
- Some metrics in Sec.2 are a bit confusing to me. Please see the questions.
- The RQ1 is valuable but not that interesting. Many papers have shown that merging techniques will lead to performance degradation.

**Questions:**

- The definition of some metrics in Sec.2 is unclear.
    - Regarding the parameter sign change ratio in Sec.2, under what situation would this be an issue? For example, consider a two-dimensional space where one weight vector is $[1, 0]$ and the other is $[-1, 1]$, then merging them leads to a vector in-between. Since the ratio is 50% in this case, the author would think it is harmful but in my opinion, I dont know why this is an issue. Could you please give more explanation?
    - For the parameter magnitude change ratio, if the task vectors are $[1, 1]$ and $[2, 2]$, this ratio could be large, but I think there is not an issue as well.
- There are some new merging techniques that the author can discuss [1-4] (maybe after the rebuttal phase).
- [5] studies the generalization bound of model merging. The author may discuss it in this paper.

[1] Stoica, George, et al. "Model merging with svd to tie the knots." *arXiv preprint arXiv:2410.19735* (2024).

[2] Zhang, Haobo, and Jiayu Zhou. "Unraveling LoRA Interference: Orthogonal Subspaces for Robust Model Merging." *arXiv preprint arXiv:2505.22934* (2025).

[3] Huang, Chenyu, et al. "Emr-merging: Tuning-free high-performance model merging." *Advances in Neural Information Processing Systems* 37 (2024): 122741-122769.

[4] Tang, Anke, et al. "Merging models on the fly without retraining: A sequential approach to scalable continual model merging." *arXiv preprint arXiv:2501.09522* (2025).

[5] Li, Hongkang, et al. "When is task vector provably effective for model editing? a generalization analysis of nonlinear transformers." *arXiv preprint arXiv:2504.10957* (2025).

---

### Official Review · Reviewer_Wxos · 2025-10-31

**Soundness:** 2
**Presentation:** 2
**Contribution:** 2
**Rating:** 4
**Confidence:** 3

**Summary:**

The paper investigates task-level collapse in model merging and shows through broad experiments across methods and backbones that the phenomenon is widespread and largely driven by the specific task rather than the merging rule. It then offers a rate–distortion perspective with a dimension-dependent lower bound on representation distortion, explaining why even convex merges can fail under realistic separations of task representations. Finally, it introduces HiddenSim and its multi-task extension MDS to quantify representation similarity, demonstrating strong correlation with merging loss and practical value for screening incompatible task combinations.

**Strengths:**

1. Breadth of evidence for collapse. Systematic experiments across multiple merging rules and backbones show the collapse phenomenon is widespread and primarily task-driven rather than method-driven.

2. Representation-space explanation that matches practice. A rate–distortion analysis yields a dimension-dependent lower bound on representation distortion for convex merges, providing a principled reason merges can fail and aligning with the empirical signal.

3. Actionable pre-screening tools. HiddenSim and its multi-task extension MDS correlate strongly with merging loss and enable swapping out incompatible tasks before merging.

**Weaknesses:**

1. Theory relies on strong assumptions. The lower-bound analysis assumes linear-mode connectivity and last-layer linearity.

2. Narrow design of HiddenSim. It uses few samples and only the last layer with L2 distance. Including ablations over sample size, layer choice, pooling, and alternative metrics would help.

3. Results are centered on GLUE and Lots-of-LoRAs. The authors may include non-classification or code/generation tasks and additional model backbones to test generality.

**Questions:**

See weakness. I'm willing to increase the rating once my concerns are well addressed.

---

### Official Review · Reviewer_X3wK · 2025-10-31

**Soundness:** 2
**Presentation:** 2
**Contribution:** 3
**Rating:** 4
**Confidence:** 3

**Summary:**

The paper investigates the failure of model merging, which occurs due to catastrophic performance degradation after merging. The authors focus their experiments on GLUE tasks and Lots-of-LoRAs checkpoints.
They define merging loss as the relative difference between the performance of the merged model and that of the fine-tuned model on the same task.
The study explores whether merging collapse is more correlated with the tasks or with the merging techniques, finding a statistically significant correlation with the tasks.
They further assess the relationship between merging loss and four metrics that capture parameter update conflicts (Parameter Magnitude Change Ratio, Parameter Sign Change Ratio, Conflicting Parameter Magnitude Change Ratio, and Average Cosine Similarity) and find no statistically significant correlation with any of them.
Instead, they discover that merging loss is correlated with a new metric they introduce, called Hidden State Distance Similarity, which measures the distance between hidden states across tasks.
Finally, they prove a theorem showing that, under the assumption of linear mode connectivity, the minimum achievable hidden-state distortion is bounded.

**Strengths:**

- The paper provides a well-structured empirical investigation of model merging failures. By systematically testing correlations between merging loss, task identity, and several parameter-space metrics, it offers strong evidence that task characteristics primarily drive performance degradation.
- The paper introduced a novel metric which  connects merging performance to the geometry of model representations and correlates with performance degradation in merging.
- The provide a theoretical result that adds a principled foundation to its empirical findings.

**Weaknesses:**

- The paper does not engage with existing literature tackling the same issue from complementary angles. Works such as for intance Task Singular Vectors [1], and Iso-C [2] directly analyze model merging collapse through task subspace geometry and rank alignment, yet are not cited or discussed. This omission limits the contextualization of the proposed study and leaves unclear how it connects to or extends prior understanding of merging failure.
- Evaluating the hidden-state diameter $\Delta$ or the distortion $\delta_{max}$ requires access to hidden representations across all fine-tuned models and the entire input distribution, which is generally infeasible in realistic multi-task or large-scale settings.
Thus, although the bound provides theoretical insight into when merging is doomed to fail, it is difficult to operationalize as a predictive or diagnostic tool.
- The statement of the theorem is not clear; some symbols are not well explained beforehand.
The symbol $D$ (and $D^*$) is introduced without a clear definition.
It appears to correspond to some notion of ``distortion'' or hidden-state mismatch, but no explicit mathematical formulation is given.
Likewise, the term distortion is used interchangeably with $\delta_{\max}(\hat{\theta})$ in the main theorem, but this identification is never formally stated.
In the proof, $\delta_i(\hat{\theta})$ is not defined; I assumed it was meant to be
$
\delta_i(\hat{\theta}) = \mathbb{E}_X \big\| H_i(X) - h(X; \hat{\theta}) \big\|_2^2 .
$
Moreover, the proof contains some small issues in Step~1: the paper's wording makes it sound as if the coefficients $\alpha$ are fixed \emph{a priori} and then one finds the center of the convex hull.
Under this interpretation, it is not true that $h(x; \bar{\theta})$ coincides with $c(x)$.
What is true is that, by Jung's Theorem, there exists a set of coefficients (depending on $x$) such that the convex combination (that is, the center) minimizes the worst-case distance to all $H_i(x)$.
If this same set of coefficients defines a parameter merge, then this merge achieves the minimum possible worst-case hidden-state distortion.
In any case, the statements appear correct to me.
- table 7-10 are not clear, what are the columns in this tables? Do they correspond to the task of one group? Moreover are never mentioned in the paper.

Typos:
- Line 317 starts with a comma.
- Line 063 missing $\mathbb{R}^d$

[1]Gargiulo, Antonio Andrea, et al. "Task singular vectors: Reducing task interference in model merging." Proceedings of the Computer Vision and Pattern Recognition Conference. 2025.

[2] Marczak, Daniel, et al. "No Task Left Behind: Isotropic Model Merging with Common and Task-Specific Subspaces." Forty-second International Conference on Machine Learning.

**Questions:**

- Table 1, why merging losses have posivite values= even if merging loss was define as negative?
- What are the  exact 25 groups of 8 checkpoint IDs used in the paper?

---

### Official Review · Reviewer_8Fp6 · 2025-11-04

**Soundness:** 3
**Presentation:** 3
**Contribution:** 2
**Rating:** 2
**Confidence:** 4

**Summary:**

The paper studies merging collapse where merging task specialist models causes performance degradation. They find that some tasks are incompatible to merging where they fail irrespective of the merging method. They also find that representational incompatibility, especially measured through hidden states similarity correlates well with merging collapse.

**Strengths:**

Strenghts:
- Good study across merging methods and domains
- Concretely find that task incompatibility causes merging collapse
- Propose a hidden similarity metric for guiding merging

**Weaknesses:**

- There is no significant understanding on merging collapse as proposed in abstract. It is known that task incompatilibity causes merging failure, as it cannot maintain linear mode connectivity.
- Merging is interesting in generalization perspective, the paper doesn’t study anything related to that. If the goal is to only to retain performance of existing models, then even past works have seen that it is not possible to always retain complete performance i.e, merging collapse will happen.
- The paper finds that some tasks are not possible to merge, but is it stil true at scale? Studying this phenomenon would more your finding more concrete.
- Using hidden state similarity is limited because of it assumes access to datasets that created the experts in the first place. If you have access the experts’ data, we can train a multitask model that performs well on all tasks instead of merging them.

**Questions:**

How does MDS metric guide merging? Do you rank the experts based on the merged model and current expert?

---

### Meta-Review · Area_Chair_R4PV · 2026-01-01

**Summary:**

This paper investigates the phenomenon of task-level model-merging collapse, where merging independently fine-tuned task specialists consistently leads to severe performance degradation. Through extensive experiments across multiple merging methods, tasks, and model settings, the authors show that collapse is largely task-driven rather than method-dependent. They further demonstrate that representational incompatibility, measured via hidden-state similarity, correlates strongly with merging loss, whereas several parameter-space conflict metrics do not. To complement the empirical findings, the paper presents a rate–distortion–based theoretical analysis that derives a lower bound on representation distortion under convex merging, aiming to explain why merging can fundamentally fail for certain task combinations.

The main strength of the paper lies in its systematic empirical evaluation. The authors conduct broad experiments across different merging techniques, backbones, and task combinations, providing convincing evidence that merging collapse is a stable and widespread phenomenon driven primarily by task characteristics. The careful correlation analysis helps disentangle task effects from method effects, and the finding that representation-space metrics are more informative than parameter-space conflicts offers a useful perspective for understanding merging failures. Additionally, the attempt to ground the empirical observations in a theoretical rate–distortion framework is well motivated and adds conceptual clarity to why representational incompatibility can impose fundamental limits on mergeability.

Despite these strengths, the paper’s incremental contribution over existing work is limited. The central claim that task incompatibility leads to merging failure aligns closely with prior intuitions and recent studies on task subspaces and representation alignment, yet the paper does not sufficiently engage with or differentiate itself from this literature. As a result, the novelty of the empirical findings is somewhat unclear. The proposed HiddenSim/MDS metrics rely on access to hidden states and task data, which may be unrealistic in large-scale or practical merging scenarios, limiting their applicability as diagnostic or prescreening tools. The theoretical analysis depends on strong assumptions (e.g., linear mode connectivity and linearized layers), and parts of the presentation lack clarity in definitions and notation, reducing the rigor and interpretability of the results. Moreover, the experimental scope remains largely confined to GLUE-style classification tasks and LoRA-based settings, leaving questions about generality to other task types and architectures.

Overall, this paper provides a solid and well-executed empirical study of task-level merging collapse and offers a plausible representational explanation supported by theory. However, the core insights are closely aligned with existing understanding, the theoretical contribution is primarily explanatory rather than predictive, and the practical impact of the proposed metrics is uncertain. Given the limited novelty and remaining concerns about scope and rigor, I lean toward reject at this time.

**Reviewer Concerns:**

The rebuttal addresses a subset of the reviewers’ concerns related to clarification and presentation. In particular, it provides additional explanations of the hidden state similarity metrics, including HiddenSim and MDS, and clarifies that these metrics are intended as diagnostic tools rather than as replacements for multitask training. The authors also respond to several questions about experimental details, table interpretations, and metric definitions, which helps reduce some confusion noted by the reviewers. To some extent, the rebuttal reinforces the empirical claim that task level collapse is consistent across merging methods, aligning with reviewers who found the experimental evidence generally solid.

However, several key concerns remain unresolved. Most notably, the rebuttal does not sufficiently address criticisms regarding novelty and positioning with respect to prior work. The relationship to recent studies that analyze merging collapse through task subspaces, rank alignment, or representation geometry is still underdeveloped, leaving the incremental contribution unclear. Concerns about the practicality and scalability of HiddenSim and MDS persist, as the rebuttal does not provide concrete evidence that these metrics can be applied in realistic large scale settings without access to task data or hidden representations. The theoretical analysis continues to rely on strong assumptions, and while the authors defend these assumptions, the rebuttal does not significantly improve the rigor, clarity, or operational relevance of the theory. Finally, concerns about generalization beyond GLUE style classification tasks, LoRA based settings, and a narrow set of architectures remain outstanding, as no new experimental results are offered to support broader claims.

**Reviewer Scores:**

Reviewer 8Fp6 would likely keep their reject score, as their main concerns about limited novelty and practical impact are not substantially addressed by the rebuttal.

Reviewer X3wK might slightly improve their confidence in the empirical results, but the unresolved issues on related work, theory clarity, and generalization suggest the score would remain marginally below the acceptance threshold.

Reviewer Wxos could view the clarifications positively, yet the key limitations in theory assumptions and experimental scope persist, so the score would likely stay marginally below threshold.

Reviewer sAmL is unlikely to change their marginally above threshold score, and could even slightly lower it given that broader scope and novelty concerns remain unaddressed.

---

### Decision · Program_Chairs · 2026-01-26

Reject